# Local translatome sustains synaptic function in impaired Wallerian degeneration

Maria Paglione[1,3], Leonardo Restivo [1], Sarah Zakhia [2], Arnau Llobet Rosell [1], Marco Terenzio [2] & Lukas J Neukomm [1✉]

## Abstract

After injury, severed axons separated from their somas activate programmed axon degeneration, a conserved pathway to initiate their degeneration within a day. Conversely, severed projections deficient in programmed axon degeneration remain morphologically preserved with functional synapses for weeks to months after axotomy. How this synaptic function is sustained remains currently unknown. Here, we show that dNmnat overexpression attenuates programmed axon degeneration in distinct neuronal populations. Severed projections remain morphologically preserved for weeks. When evoked, they elicit a postsynaptic behavior, a readout for preserved synaptic function. We used ribosomal pulldown to isolate the translatome from these projections 1 week after axotomy. Translatome candidates of enriched biological classes identified by transcriptional profiling are validated in a screen using a novel automated system to detect evoked antennal grooming as a proxy for preserved synaptic function. RNAi-mediated knockdown reveals that transcripts of the mTORC1 pathway, a mediator of protein synthesis, and of candidate genes involved in protein ubiquitination and Ca²⁺ homeostasis are required for preserved synaptic function. Our translatome dataset also uncovers several uncharacterized Drosophila genes associated with human disease. It may offer insights into novel avenues for therapeutic treatments.

**Keywords** Wallerian Degeneration; Programmed Axon Degeneration; Local mRNA Translation; Synaptic Function; Antennal Grooming Behavior
**Subject Categories** Neuroscience; Translation & Protein Quality

## Introduction

Wallerian degeneration is a simple and well-established system to investigate how damaged axons execute their destruction (Waller, 1850). Upon axonal injury (axotomy), the axon separated from the soma employs programmed axon degeneration to initiate its degeneration within a day. The resulting axonal debris is cleared by phagocytosis through neighboring glial cells in subsequent days (Raiders et al, 2021; Sapar and Han, 2019). Programmed axon degeneration is conserved among various species and appears to be hijacked without axotomy in several neurological conditions (Coleman and Höke, 2020; Llobet Rosell and Neukomm, 2019).

In *Drosophila*, so far, programmed axon degeneration is mediated by four genes and a single metabolite (Fang et al, 2012; Llobet Rosell et al, 2022; Neukomm et al, 2017; Osterloh et al, 2012; Paglione et al, 2020; Xiong et al, 2012). *Drosophila* nicotinamide mononucleotide adenylyltransferase (dNmnat) is synthesized in the soma and transported into the axon, where it is degraded by the E3 ubiquitin ligase Highwire (Hiw). Axonal transport and degradation result in steady-state dNmnat, which consumes nicotinamide mononucleotide (NMN) to generate nicotinamide adenine dinucleotide (NAD⁺) in an ATP-dependent manner. Upon axotomy, the axonal transport of dNmnat is abolished. Consequently, dNmnat rapidly decreases together with NMN consumption and NAD⁺ synthesis. The resulting increase of NMN activates *Drosophila* Sterile Alpha and TIR Motif-containing protein (dSarm), a NADase that pathologically depletes axonal NAD⁺. Low NAD⁺ results in the degeneration of the injured axon mediated by Axundead (Axed).

The manipulation of programmed axon degeneration results in severed axons and associated synapses (projections) that remain morphologically preserved for weeks to months. When evoked, they elicit a postsynaptic behavior, suggesting that synapses remain functionally preserved. In *hiw* mutant larvae, severed projections continue to elicit evoked excitatory junction potentials (EJPs) and spontaneous mini EJPs (mEJPs) in muscles up to 24 h after axotomy (Xiong et al, 2012). In diverse programmed axon degeneration adult mutants, the evoked stimulation of severed antennal grooming-inducing sensory neuron projections results in behavior for at least 2 weeks after axotomy (Llobet Rosell et al, 2022; Neukomm et al, 2017; Paglione et al, 2020). It is also observed in mice, where muscle fibers respond to evoked severed motor projections for up to 5 days after axotomy (Mack et al, 2001). Thus, attenuated programmed axon degeneration preserves morphology and synaptic function in severed projections for weeks after axotomy.

The metabolite NAD⁺ is instrumental in the morphological preservation of severed projections. While sustained NAD⁺ levels ensure preservation, its forced depletion triggers rapid axon and neurodegeneration (Essuman et al, 2017; Gerdts et al, 2015;

[1]Department of Fundamental Neurosciences, University of Lausanne, 1005 Lausanne, Switzerland. [2]Molecular Neuroscience Unit, Okinawa Institute of Science and Technology Graduate University, Kunigami-gun, Okinawa 904-0412, Japan. [3]Present address: Lemanic Neuroscience Doctoral School (LNDS), Lausanne, Switzerland. ✉E-mail: lukas.neukomm@unil.ch

Neukomm et al, 2017). However, the mechanisms ensuring the preservation of synaptic function are currently unknown.

In mice with attenuated programmed axon degeneration (Wallerian degeneration slow, $Wld^S$), severed projections contain increased numbers of polyribosomes packed in multimembrane vesicles in the neurofilament space, suggesting that local protein synthesis may contribute to sustaining axonal homeostasis (Court et al, 2008). However, applying cycloheximide or emetine as protein synthesis inhibitors does not alter the morphological preservation after axotomy (Gilley and Coleman, 2010). Because local protein synthesis is crucial for synaptic plasticity (Holt et al, 2019; Ostroff et al, 2019; Yoon et al, 2012), we hypothesized that in severed projections, it contributes to the preservation of synaptic function.

Here, we use Drosophila to demonstrate that dNmnat-mediated overexpression ($dnmnat^{OE}$) potently attenuates programmed axon degeneration for weeks after axotomy. Consequently, severed projections of distinct sensory neuron populations remain morphologically preserved and their synapses functional. We employed tissue-specific ribosome pulldowns to isolate translated transcripts 1 week after axotomy. In-depth transcriptional profiling revealed several enriched biological classes. To validate our dataset, we established a novel system that automatically detects and quantifies antennal grooming behavior as a proxy for preserved synaptic function. We used this system to perform a high-throughput RNAi-mediated screen, which led to the identification of several protein ubiquitination and $Ca^{2+}$ homeostasis candidates, and genes of the mTORC1-mediated protein synthesis pathway. Our observations demonstrate that local protein synthesis is required to preserve synaptic function in a model of impaired Wallerian degeneration.

# Results

## Preserved axonal morphology and synaptic function in severed projections with $dnmnat^{OE}$-mediated attenuated programmed axon degeneration

We used a Drosophila model of impaired Wallerian degeneration to isolate and explore the axonal and synaptic translatome (Llobet Rosell and Neukomm, 2019). To attenuate programmed axon degeneration and isolate the local translatome, we decided against programmed axon degeneration mutants, which are homozygous lethal. They require Mosaic Analyses with a Repressible Cell Marker (MARCM); thus, the local translatome can only be isolated in sparse clonal neurons (Lee and Luo, 1999; Neukomm et al, 2014). We used the Gal4/UAS system to over-express dNmnat ($dnmnat^{OE}$) in neuronal tissues. However, in previous attempts, dNmnat overexpression only partially attenuated programmed axon degeneration (MacDonald et al, 2006).

To test the $dnmant^{OE}$ preservation, we used an untagged cytoplasmatic dnmnat isoform in $dpr1^+$ sensory neuron clones in the wing (Zhai et al, 2006) (Table EV1). In 7-day-old animals, one wing was subjected to partial axotomy, while the other served as an uninjured control. We quantified GFP-labeled injured and uninjured control axons at 20 and 25 °C, 7 and 14 days post axotomy (dpa). While wild-type axons degenerated within days after axotomy, robust $dnmnat^{OE}$-mediated preservation was observed up to 14 dpa (Fig. 1A,B). The preservation appeared stronger at 25 °C; we thus conducted subsequent experiments at this temperature.

Next, we used $dnmnat^{OE}$ in ~20 $or22a^+$ GFP-labeled olfactory receptor neurons (ORN). Their cell bodies are housed in 3rd antennal segments, and their axons project into the antennal lobe (Vosshall et al, 2000) (Fig. 1C). The bilateral ablation of 3rd antennal segments results in the removal of the cell bodies and the degeneration of ipsi- and contralateral projections (MacDonald et al, 2006). While wild-type axons degenerated after axotomy, severed $dnmnat^{OE}$ projections remained strongly preserved at 7 and 14 dpa (Fig. 1C). Quantifying $or22a^+$ axons and their projection area as a proxy for the synaptic field confirmed the strong $dnmnat^{OE}$-mediated preservation (Fig. 1D). We made similar observations with approximately 40 $JO^+$ mechano-sensory neurons in the Johnston's Organ (JO). $JO^+$ cell bodies are housed in 2nd antennal segments and project their axons into the subesophageal zone (Hampel et al, 2017, 2015) (Fig. 1E). While the bilateral ablation of 2nd antennal segments resulted in the complete degeneration of $JO^+$ wild-type projections, $dnmnat^{OE}$ projections remained preserved at 14 dpa (Fig. 1E,F). Our observations suggest that $dnmnat^{OE}$ potently blocks the activation of programmed axon degeneration signaling, thereby preserving the morphology of severed projections in various neurons for weeks after axotomy.

We used the $JO^+$ neurons, which are required and sufficient for antennal grooming, to assess the preservation of synaptic function (Hampel et al, 2015, 2017; Neukomm et al, 2017; Paglione et al, 2020). First, we determined the age of adult animals where evoked antennal grooming behavior is consistently robust. Animals were fed with all-trans retinal during development, and adults expressing a red-shifted Channelrhodopsin (CsChrimson) in $JO^+$ neurons were subjected to three consecutive 10 s 10 Hz red-light exposures (see Methods for details). Independent groups of adult animals were exposed once to red light from 0–1 to 7–8 days post eclosion (dpe). We observed a robust evoked antennal grooming behavior at 7–8 dpe and used this age for subsequent experiments (Fig. EV1).

We applied the same axotomy paradigm described above in manipulated $JO^+$ neurons. While optogenetics failed to evoke antennal grooming in wild-type animals at 7 dpa, animals with $dnmnat^{OE}$ $JO^+$ neurons continued to elicit antennal grooming (Fig. 1G). We made a similar observation at 14 dpa, albeit with a significantly reduced evoked behavior (Fig. 1G). Our observation suggests that in severed $dnmnat^{OE}$ projections, synapses remain functional for at least 1 week after axotomy.

The abundance of protein synthesis is lower in dendrites and axons than in the soma (Glock et al, 2021). To increase the biological material, we sought a Gal4 driver that labels many or all sensory neurons whose cell bodies can be readily ablated. We identified the pan-ORN Gal4 driver olfactory receptor co-receptor (orco–Gal4) that is transcriptionally active in approximately 800 ORNs (Vosshall et al, 2000; Wang et al, 2003). $orco^+$ cell bodies are housed in antennae and maxillary palps, and their axons project into the antennal lobe (Fig. 1H). Bilateral antennal and maxillary palp ablation resulted in the degeneration and debris clearance of wild-type axons at 14 dpa. In contrast, severed $dnmnat^{OE}$ projections remained extensively preserved (Fig. 1H,I). These results demonstrate that $dnmnat^{OE}$ prevents axonal and synaptic degeneration for weeks after axotomy, regardless of the number of manipulated neurons. In subsequent experiments, we used 7 dpa to isolate the local translatome from severed projections, as morphology and synaptic function remained unchanged.

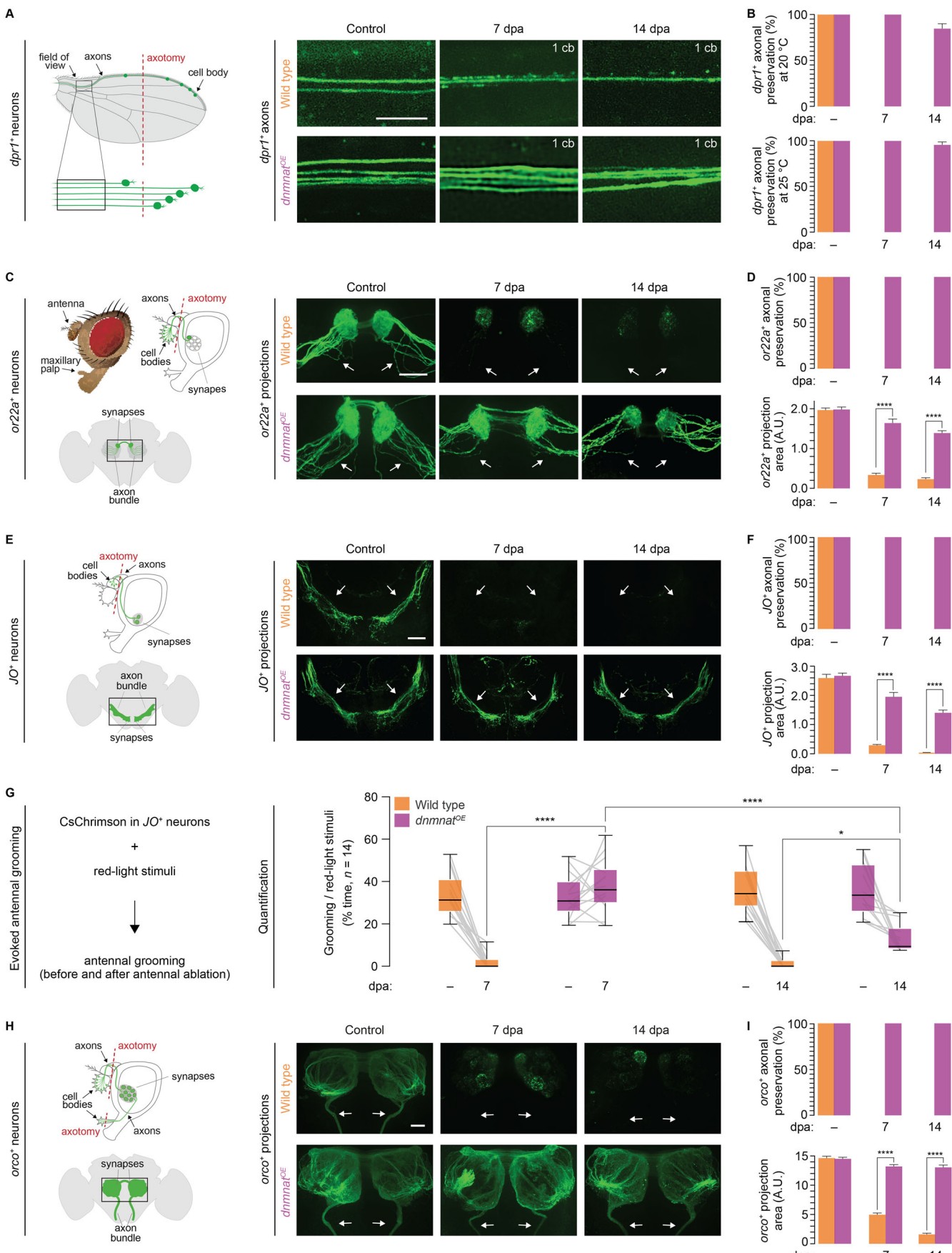

**Figure 1.  *dnmnat^OE*-mediated attenuated programmed axon degeneration preserves axons and synapses for weeks after axotomy.**

(A) Left, schematic wing with sparsely GFP-labeled *drp1*+ sensory neuron MARCM clones. Axotomy and field of observation are indicated. Right, examples of *drp1*+ GFP-labeled wing axonal projections. Wild type and overexpression of dNmnat (*dnmnat^OE*), top and bottom; uninjured (control), and axotomized (7 and 14 days post axotomy (dpa)), respectively. Scale bar, 10 μm. The number of remaining cell bodies (cb), and therefore intact axons, is indicated in the upper right corner of each example. (B) Quantification of intact *drp1*+ axons at 20 and 25 °C; top and bottom, respectively. Data, average ± SEM (*n* = 15 wings). (C) Left, schematic head side and brain front view with indicated *or22a*+ GFP-labeled cell bodies, axons, and synapses. Right, examples of *or22a*+ GFP-labeled olfactory receptor neuron axonal projections (arrow). Wild type and *dnmnat^OE*, top and bottom; control, and 7 and 14 dpa, respectively. Scale bar, 20 μm. (D) Quantification of intact *or22a*+ axons and total axonal and synaptic area; top and bottom, respectively. Data, average ± SEM (*n* ≥ 10 brains). Unpaired two-tailed t-student test. (E) Left, schematic head side and brain front view with *JO*+ GFP-labeled cell bodies, axons, and synapses. Right, examples of GFP-labeled *JO*+ mechanosensory axonal projections (arrow). Wild type and *dnmnat^OE*; control, and 7 and 14 dpa, respectively. Scale bar, 20 μm. (F) Quantification of intact *JO*+ axons and total axonal and synaptic area. Data, average ± SEM (*n* ≥ 10 brains). Unpaired two-tailed t-student test. (G) Robust preservation of synaptic function in severed *dnmnat^OE* projections at 7 dpa. Left, schematic of optogenetic-induced antennal grooming behavior. Right, quantification of manually scored antennal grooming in uninjured and injured animals. Data, % time of red-light stimuli (*n* = 14 animals); box (interquartile) and whisker (minimum and maximum, respectively) plot, with minimum, lower quartile (Q1), median, upper quartile (Q3), and maximum. Three-way ANOVA with Tukey's multiple comparisons test. (H) Left schematic head side and brain front view with *orco*+ GFP-labeled cell bodies, axons, and synapses. Right, examples of olfactory receptor co-receptor (*orco*+) GFP-labeled wild-type and *dnmnat^OE* projections (arrow). Wild type and *dnmnat^OE*; control, and 7 and 14 dpa, respectively. Scale bar, 20 μm. (I) Quantification of *orco*+ intact axons and projection area, respectively. Data, average ± SEM (*n* ≥ 5 brains). Unpaired two-tailed t-student test. ****p < 0.0001; *p < 0.05. Source data are available online for this figure.

## Local axonal and synaptic translatomes of severed, preserved projections 1 week after axotomy

We sought to test whether the GFP-tagged large ribosomal subunit protein RpL10Ab (GFP::RpL10Ab) can be detected and thus used for translating ribosome affinity purification (TRAP) in severed projections (Thomas et al, 2012). While the Gal4/UAS-mediated expression of GFP::RpL10Ab was neither detected in 20 *or22a*+ nor 40 *JO*+ neurons, 800 *orco*+ neurons revealed a GFP signal in Western blots derived from heads (Fig. EV2). Thus, *orco*+ neurons were used for tissue-specific TRAP (Fig. 2A). Briefly, GFP::RpL10Ab was expressed in *orco*+ wild-type and *dnmnat^OE* olfactory organs. Both genotypes were subjected to antennal and maxillary palp ablations, and at 7 dpa, 300 heads were collected and snap-frozen each. Subsequently, axonal and synaptic-specific ribosomes were immunoprecipitated with anti-GFP antibody-coated magnetic beads, mRNAs extracted, and reverse transcribed to establish cDNA libraries (Fig. 2A, Methods).

We identified 1033 differentially enriched axonal and synaptic transcripts in the two genotypes at 7 dpa, with 499 transcripts enriched in *dnmnat^OE* (Fig. 2B, Dataset EV1, EV7, EV8). These enriched transcripts, with a regularized logarithm (rLog) fold change value > 0.6 and a false discovery rate (FDR) < 0.05, were run through the database for annotation, visualization, and integrated discovery (DAVID) for gene ontology (GO) analyses of biological processes (Fig. 2C, Dataset EV2). Notably, we observed a significant transcript enrichment of Ca²⁺ transport (*p* = 0.00017), axon extension (*p* = 0.0004), axon guidance (*p* = 0.002), RNA processing (*p* = 0.007), as well as oxidation/reduction, ubiquitination, and transmembrane transport (*p* = 0.03 each, respectively) (Fig. 2C). Similar observations in primary *Nmnat2*⁻/⁻ cortical neurons were previously reported, however, with an inversed altered transcriptional profile (Niou et al, 2022).

Next, we looked at the expression levels (e.g., rLog values) of our 499 identified GO term class transcripts (Fig. 2C) in the distinct conditions and genotypes (e.g., control, 7 dpa, wild type, and *dnmnat^OE*, respectively) (see Methods for details). Interestingly, protein ubiquitination transcripts were the sole GO term class enriched in *dnmnat^OE* solely after axotomy (Fig. 2D, Dataset EV2, EV7). In contrast, transcripts of Ca²⁺ transport were already enriched before axotomy, suggesting a potential transcriptional response to *dnmnat^OE* (Fig. 2D).

We made a similar observation with transcripts related to axon guidance and extension, G-protein receptors, transmembrane transport, vesicle-mediated transport, autophagy, RNA processing, and 50% of the transcripts of oxidative stress (Fig. EV3, Dataset EV2, EV7). Our analyses revealed enriched, *dnmnat^OE*-independent and dependent transcripts of distinct biological GO terms.

A recent study examined in vivo mRNA decay in mammals by analyzing the axonal transcriptome in severed axons of *Sarm1*⁻/⁻ mice (Jung et al, 2023). mRNAs were collected from *Sarm1*⁻/⁻ retinal ganglion cell (RGC) axons at 7 dpa and compared to naive (uninjured) axons of the same genotype. Notably, while *Sarm1*⁻/⁻ mice have a different genotype than *dnmnat^OE* flies, both genotypes prevent the activation of programmed axon degeneration after axotomy (Fang et al, 2012; Gilley and Coleman, 2010; Osterloh et al, 2012). The comparison of the two 7 dpa datasets (*Sarm1*⁻/⁻ axonal transcriptome vs. *dnmnat^OE* TRAP translatome) revealed 61 enriched, orthologous mouse/*Drosophila* transcripts in both datasets (Fig. 2E). Biological GO term analyses revealed an enrichment of transcripts involved in protein polyubiquitination (*p* = 0.009), Ca²⁺ transmembrane transport (*p* = 0.008), and transmembrane transport (*p* = 0.042) (Fig. 2F, Dataset EV3). Therefore, our analysis of mouse and *Drosophila* orthologs revealed a common set of enriched biological GO term transcripts after axotomy.

We also implemented an additional control using TRAP-seq from *orco–Gal4* alone (background) to further restrict our analyses (Fig. 3A). Both wild-type and *dnmnat^OE* samples were filtered against this background (Fig. 3B–D, Dataset EV1, EV4, EV5, EV7). The additional stringent filtering led to the identification of 165 transcripts in *dnmnat^OE* at 7 dpa (Fig. 3E). However, the biological GO term analyses of both strategies showed that most transcripts were similar to those found in the initial dataset (Figs. 2F, 3F, Dataset EV3, EV6), where protein ubiquitination and Ca²⁺ transport appeared significantly enriched in the local translatome datasets.

## A robust automated and high-accuracy antennal grooming detection system as a readout for preserved synaptic function

We aimed to validate whether the identified candidate transcripts are required to preserve synaptic function in severed, preserved projections 1 week after axotomy. Given the large number of

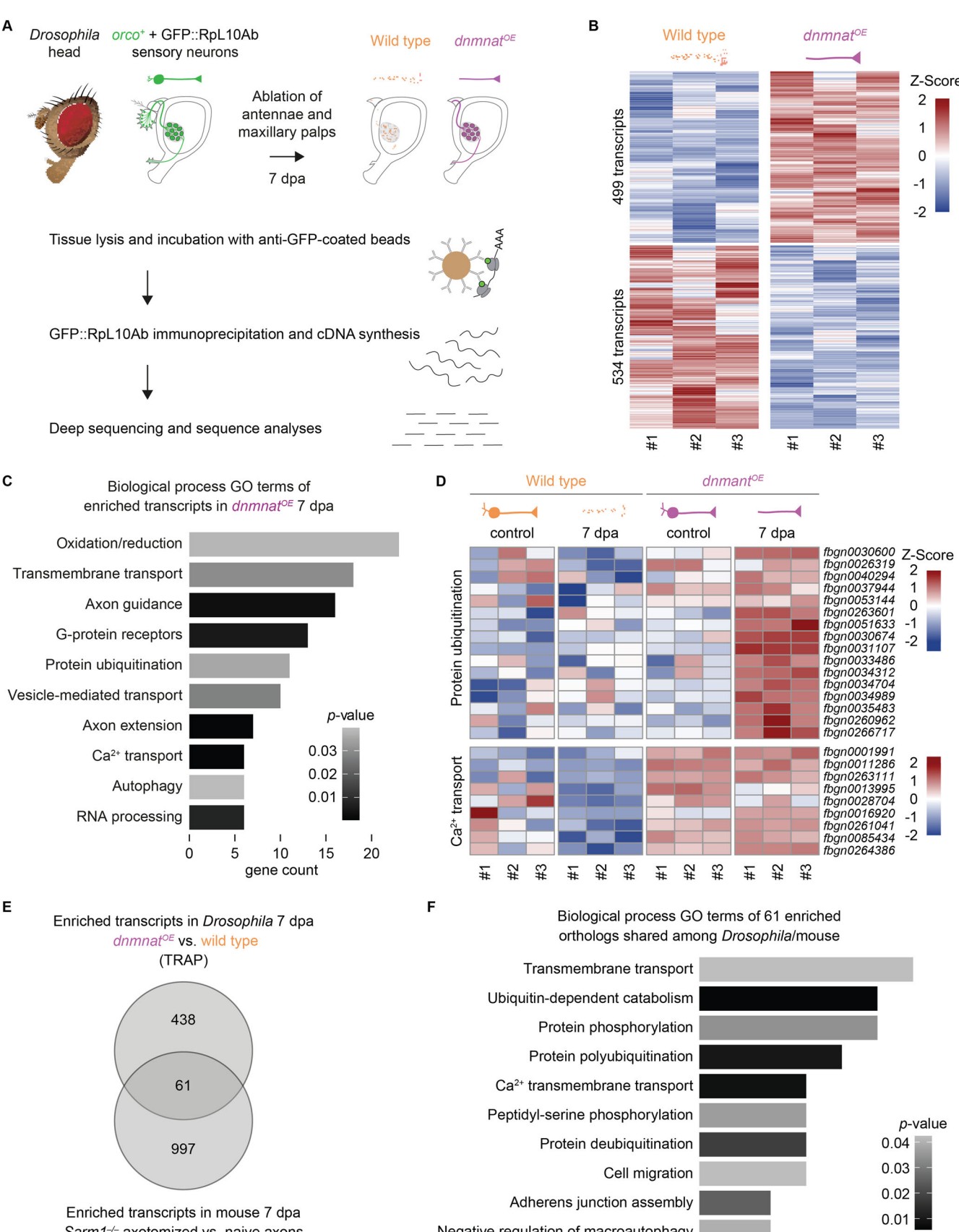

**Figure 2.  Identification of local translatomes in severed olfactory organ axons and synapses 1 week after axotomy.**

(A) Translating ribosome affinity purification (TRAP) as a strategy to isolate $orco^+$ translatomes from severed axons and their synapses 7 dpa. (B) Heat map of 1033 differentially enriched transcripts between wild type and $dnmnat^{OE}$ 7 dpa (Z-Score average) ($n = 300$ heads for each genotype and condition). (C) Gene ontology (GO) term enrichment analysis (biological processes) of 499 $dnmnat^{OE}$-enriched transcripts (fold change value > 0.6); sorted and plotted by $p$-value. Expression Analysis Systematic Explorer (EASE) score, a modified Fisher's Exact test. (D) Heat map example of two significantly changed GO term groups, protein ubiquitination ($n = 16$) and $Ca^{2+}$ transport ($n = 9$), respectively. (E) Venn diagram of 499 *Drosophila* $dnmnat^{OE}$-enriched TRAP transcripts and 1058 mouse $Sarm1^{-/-}$ KO transcripts (axonal transcriptome; axotomized versus naive axons (Jung et al, 2023)). (F) GO term analysis (biological processes) of 61 enriched *Drosophila*/mouse ortholog transcripts (fold change value > 0.6); sorted and plotted by $p$-value. Expression Analysis Systematic Explorer (EASE) score, a modified Fisher's Exact test. Source data are available online for this figure.

candidates, we decided to develop a system that rapidly detects antennal grooming behavior since manual analysis is labor-intensive.

We developed two neural networks that perform frame-wise scoring of the movement of forelegs on antennae, e.g., antennal grooming behavior. Briefly, Network 1 identifies the head region of the animal and enables the reduction of the analysis on a small area of the recording (Fig. 4A). Network 2 uses this area and overlays three frames to identify pixel-wise standard deviations to generate a compressed image for antennal grooming behavior detection (Methods) (Fig. 4A). The system was trained and validated to identify grooming or no grooming events in these compressed images, resulting in a frame-wise probability of antennal grooming (Figs. 4A and EV4A). After training, the system produced probability scores similar to the manual scores of antennal grooming made by the trained observer (Fig. 4B).

To identify the probability cutoff where the quantification difference between the probability score assigned by the system and the manual score by the observer is minimal, the mean absolute differences were plotted in a probability-dependent manner (Fig. 4C). Once the probability threshold was determined, the system was further validated by a test set. A minimal difference was observed with a probability ≥0.3 (Fig. 4C). A linear regression between system and observer scores confirmed a strong correlation with a ≥0.3 probability cutoff (Fig. 4D). Despite the >93% performance achieved by the system on the test set (Methods), we further compared the detection of antennal grooming from the system with the scores from the observer on the test set. Grooming detection was not statistically significant between the system and the observer (Fig. 4E). We therefore conclude that our newly established two-network system plots accurate probabilities of evoked antennal grooming (Movie EV1).

We employed this system primarily after antennal ablation. However, we observed that even the presence or absence of antennae has a significant impact on the performance of network 2. We used a second training set of videos where antennae were removed (e.g., 7 dpa) to identify grooming or no-grooming events in compressed images after axotomy. We repeated the same procedure with ablated antennae and determined a ≥0.4 probability cutoff where the difference between the system and the manual scores was minimal (Fig. 4F). A linear regression between system and observer scores revealed a similar strong correlation (Fig. 4G). We used newly recorded videos where antennal grooming detection was compared between wild type and $dnmnat^{OE}$ at 7 dpa with no statistical difference between the system and the observer (Fig. 4H), confirming the accuracy of the predictions (Movies EV2 and EV3).

To validate the accuracy of the system, we compared the grooming detection output of the system with manual scores obtained by two independent observers in uninjured controls and at 7 dpa. The result of the system was in line with both observers. It suggests the system produces scores in the range of the variability measured among observers (Fig. EV4B,C).

We also tested whether different experimental conditions affect grooming behavior picked up by the observer or system. We combined two distinct temperatures (20 and 25 °C, respectively) with red-light intensities (8 and 14 mW/cm², respectively). While red-light intensities did not alter the behavior, temperature-dependent nuances were identified by both the system and the observer. Animals groomed significantly more at 25 °C (Fig. EV5). Thus, the system reliably detects the behavior of evoked antennal grooming and can be used to perform unbiased screens to identify subtle behavioral changes.

## RNAi-mediated knockdown of protein ubiquitination and Ca²⁺ transport candidates alter preserved synaptic function

We used our automated grooming detection system to validate candidates from our translatome dataset for preserved synaptic function after axotomy. Highly translated mRNAs are selectively degraded in axons; thus, their pools decrease over time (Jung et al, 2023). In our paradigm, while uninjured and injured $dnmnat^{OE}$ animals harbor a robust antennal grooming behavior, the specific candidate RNAi-mediated knockdown should reduce the behavior specifically at 7 dpa (Fig. 5A).

The comparisons of the mammalian dataset with our background filtering strategy prompted us to validate protein ubiquitination and Ca²⁺ transport GO term classes (Figs. 2E,F and 3E,F). The expression of candidate genes overlapped among $orco^+$ and $JO^+$ sensory neurons, supporting our subsequent validation in $JO^+$ neurons. Due to the design of the screen, we expressed CsChrimson from the X rather than from the second chromosome, as previously done. However, the grooming behavior remained unaltered regardless of the source of CsChrimson (Fig. 1G; Appendix Fig. S1). In $dnmnat^{OE}$ $JO^+$ neurons, we targeted 9 enriched protein ubiquitination candidates by RNAi in uninjured controls and at 7 dpa. We observed three major phenotypes. First, several candidate RNAi experiments did not change evoked antennal grooming behavior in uninjured controls or at 7 dpa, e.g., $pic^{RNAi}$ (Fig. 5B; Appendix Fig. S2A). Second, reduced grooming was observed in uninjured controls and at 7 dpa, e.g., $traf4^{RNAi}$ (Fig. 5B). Third, grooming behavior was unaltered in uninjured controls but reduced at 7 dpa, e.g., $huwe1^{RNAi}$ and the uncharacterized coding

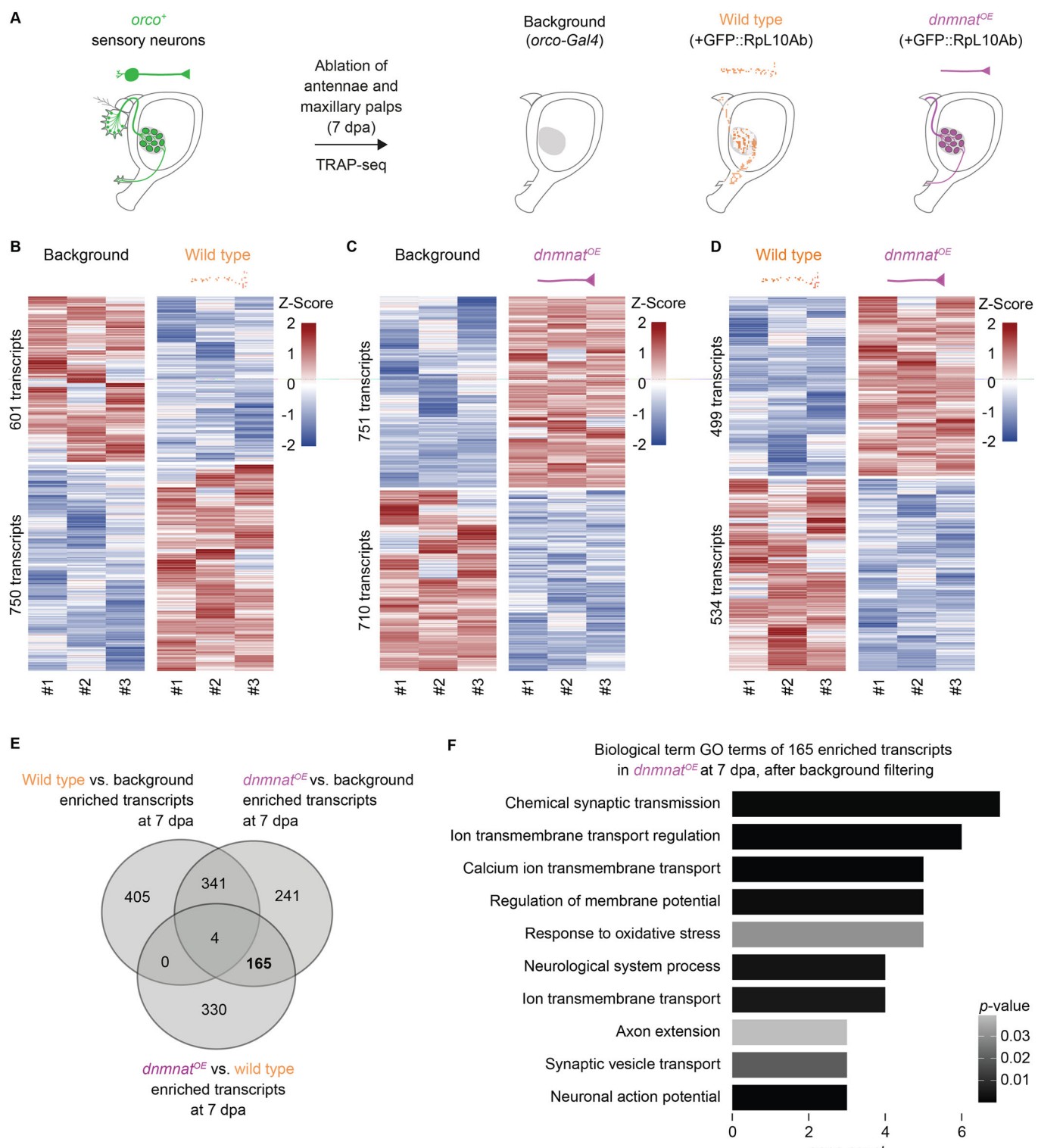

gene 10916 (*CG10916^RNAi*). RNAi-mediated knockdown of *CG6923* exhibited the most substantial decrease in evoked grooming behavior at 7 dpa (Fig. 5B). Importantly, among the three phenotypes, the morphology of the projections harbored no overt signs of degeneration (Fig. 5B; Appendix Fig. S3A). Our RNAi-mediated approach suggests that the identified transcripts by TRAP

involved in protein ubiquitination are required to sustain *dnmnat^OE*-mediated synaptic function 1 week after axotomy.

Perturbed axonal Ca²⁺ homeostasis often results in axon degeneration (Stirling and Stys, 2010). Thus, we also performed RNAi-mediated knockdown of 7 identified Ca²⁺ transport candidate transcripts. We observed similar phenotypes as described above. In *JO⁺ dnmnat^OE*

◀ **Figure 3. Background filtering of transcriptional profiles.**

(A) Strategy implemented for background filtering. (B) Heat map of 1351 transcripts differentially expressed in wild type (*orco > GFP::RpL10Ab*) compared to background (*orco–Gal4*) 7 dpa. (C) Heat map of 1461 transcripts differentially expressed in *dnmnat^OE* (*orco > GFP::RpL10Ab, dnmnat*) compared to background (*orco–Gal4*) 7 dpa. (D) Heat map of 1033 transcripts differentially expressed in *dnmnat^OE* (*orco > GFP::RpL10Ab, dnmnat)* compared to wild type (*orco > GFP::RpL10Ab*) 7 dpa. Data, Z-Score average. (E) Venn diagram of 750 transcripts enriched in wild type (compared to background), 751 transcripts enriched in *dnmnat^OE* (compared to background), and 499 transcripts enriched in *dnmnat^OE* (compared to wild type) 7 dpa. (F) Biological process GO term analysis of 165 enriched transcripts in *dnmnat^OE* after background filtering (fold change value > 0.6); sorted and plotted by *p*-value. Expression Analysis Systematic Explorer (EASE) score, a modified Fisher's Exact test. Source data are available online for this figure.

neurons, knockdown of several candidates did not alter evoked antennal grooming behavior in uninjured controls and at 7 dpa, including the pore-forming alpha[1] subunit D of the L-type voltage-gated $Ca^{2+}$ channel (*ca-alpha1D^RNAi*) (Fig. 5C; Appendix Fig. S2B). RNAi-mediated knockdown of *cacophony* (*cac^RNAi*) reduced the behavior in uninjured controls and at 7 dpa (Fig. 5C). Knockdown of the endoplasmatic reticulum ryanodine receptor (*ryr^RNAi*) required for $Ca^{2+}$ release from intra-axonal stores also decreased grooming in uninjured controls and at 7 dpa. In contrast, RNAi-mediated knockdown of the $Na^+/K^+/Ca^{2+}$ exchanger *nckx30c* (*nckx30C^RNAi*) resulted in a reduced behavior solely at 7 dpa (Fig. 5C), as observed with two different RNAi constructs (Appendix Fig. S2C). We also identified *calx^RNAi* with an increased grooming behavior solely at 7 dpa (Fig. 5C). The severed projections appeared similarly preserved compared to their controls (Fig. 5C; Appendix Fig. S3B). Our observation provides further evidence that the transcripts involved in $Ca^{2+}$ identified by TRAP are crucial for *dnmnat^OE*-mediated preservation of synaptic function.

The RNAi-based validation of the candidates identified by TRAP prompted us to ask whether a direct impairment of local translation alters synaptic function. We sought to test mTOR signaling, which is required for local axonal translation following nerve axotomy (Terenzio et al, 2018). Raptor is a component of the mTOR complex 1 (mTORC1) and regulates protein synthesis through ribosomal protein S6 kinase (S6k) (Fig. 6A). Notably, *raptor* was enriched at 7 dpa in our dataset (Fig. 2B). We performed RNAi-mediated knockdown of *raptor* and *s6k*. In *JO^+ dnmnat^OE* neurons, *raptor^RNAi* decreased behavior in uninjured controls, and at 7 dpa, we observed a statistically non-significant trend of a further decrease (Fig. 6B). In contrast, *s6k^RNAi* harbored a significantly impaired evoked antennal grooming behavior at 7 dpa (Fig. 6B). As previously observed, the severed projections remained morphologically preserved (Fig. 6B; Appendix Fig. S3C). To support the *s6k^RNAi* findings, we performed a time-course analysis where evoked antennal grooming was measured at 1, 3, 5, and 7 dpa in the same animal (Fig. 6C; Appendix Fig. S4). We observed that the evoked antennal grooming behavior of *dnmnat^OE* controls did not change over time ($R^2 = 0.06236$, with $p = 0.0742$), while *s6k^RNAi* resulted in a significant decrease ($R^2 = 0.2490$, with $p = 0.0006$). Our observations support the notion that the identified transcripts by TRAP may be locally translated in severed axons and their synapses under the control of mTOR signaling to sustain their function.

## Discussion

Here, we investigated how severed projections, with attenuated programmed axon degeneration, employ local protein synthesis to sustain synaptic function for at least 1 week after axotomy. We over-expressed *Drosophila* Nmnat (*dnmnat^OE*) to attenuate programmed axon degeneration, thus preserving axonal morphology and synaptic function after axotomy. dNmnat/NMNAT2 is evolutionarily conserved and consumes NMN as a substrate to generate $NAD^+$ in an ATP-dependent manner (Brazill et al, 2017; Llobet Rosell et al, 2022; Zhai et al, 2009, 2006).

The discovery of the Wallerian degeneration slow (*Wld^S*) mouse provides the basis for our current understanding of the preserving NMNAT function (Lunn et al, 1989). In *Wld^S*, a complex genomic rearrangement led to the fusion of the N-terminal 70 amino acids of UBE4b and full-length NMNAT1, referred to as WLD^S (Mack et al, 2001). The *Wld^S* coding gene is inserted as a triplication between *Ube4b* and *Nmnat1*. It results in over-expressed WLD^S, which is relocated from the nucleus to the axon. After axotomy, WLD^S persists in the severed axon, while endogenous NMNAT2 is rapidly degraded (Gilley and Coleman, 2010), resulting in preserved axonal morphology and synaptic function (Mack et al, 2001). Various studies confirmed the morphological preservation mediated by dNmnat in *Drosophila* (Fang et al, 2012; Llobet Rosell et al, 2022; MacDonald et al, 2006). Therefore, high levels or degradation-insensitive variants of dNmnat/NMNAT are a powerful tool for attenuating programmed axon degeneration and studying how severed projections remain preserved.

Here, we used an untagged dNmnat (Zhai et al, 2006) and observed that the morphological preservation lasts for at least 2 weeks in multiple neurons. It contrasts previous observations where the overexpression of N-terminal Myc-tagged dNmnat preserved axonal morphology for ~5 days after axotomy (MacDonald et al, 2006). This enhanced neuroprotection may be attributed to untagged dNmnat or different expression levels due to distinct transgene insertion sites.

After axotomy, in the soma-attached proximal axon, axonal regeneration is initiated through mTOR-mediated local axonal translation (Terenzio et al, 2018). More broadly, mTORC1 signaling mediates translation and tissue regeneration in axolotl compared to non-regenerative tissue in mice (Zhulyn et al, 2023). Surprisingly, severed distal *Wld^S* projections also harbor increased numbers of polyribosomes, suggesting that local protein synthesis may be in place to exert preservation (Court et al, 2008). We therefore hypothesized that severed projections engage maintenance through local protein synthesis. This appears in stark contrast with preserved *Wld^S* axonal morphology that does not depend on local translation (Gilley and Coleman, 2010). Nevertheless, we revisited this question in the context of preserved synaptic function.

Local mRNA translation in axons and synapses is less abundant than in cell bodies (Glock et al, 2021). To increase the biological

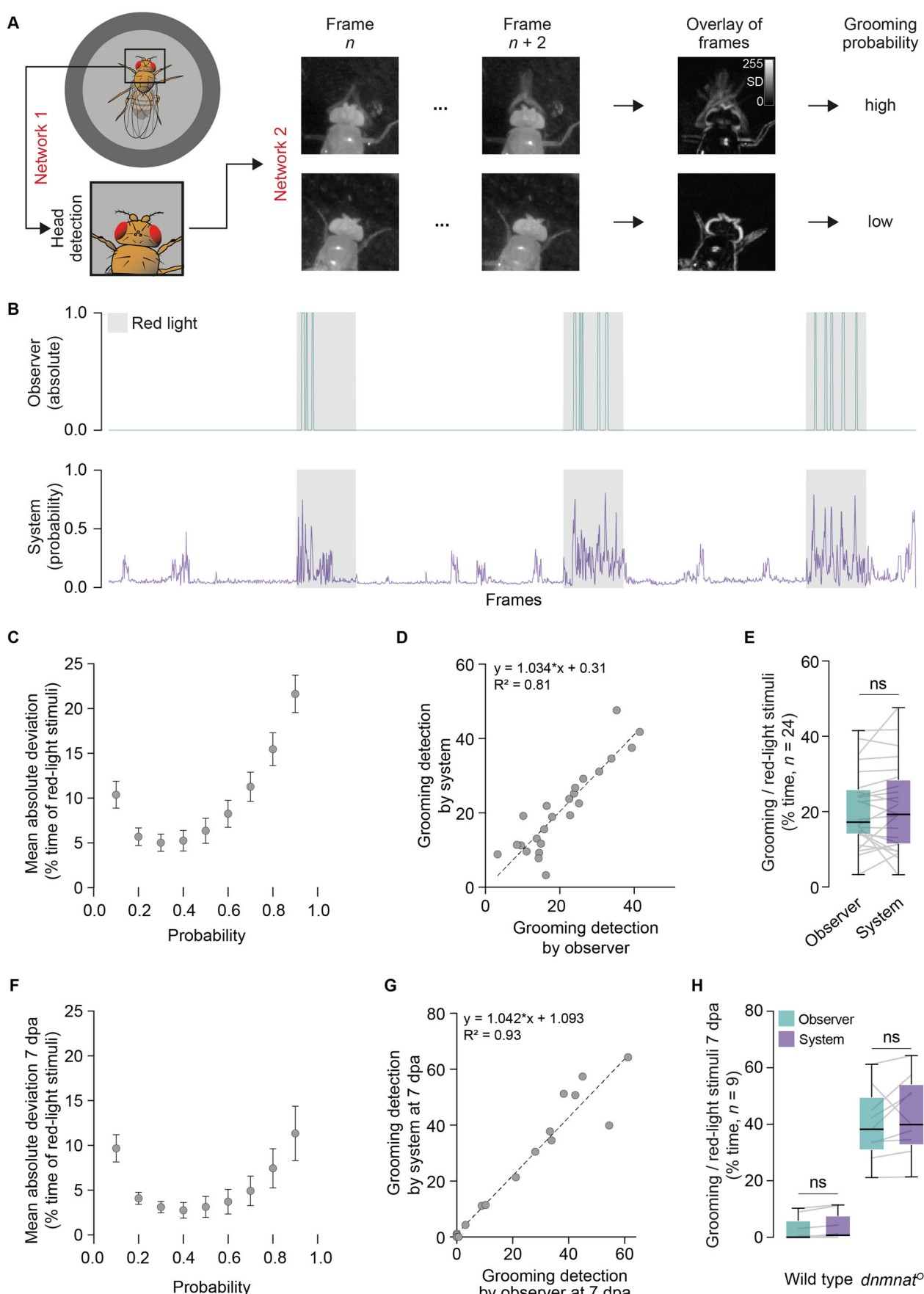

◄ **Figure 4. Automated high-accuracy antennal grooming behavior detection system.**

(A) Illustration of the two-network system for head and antennal grooming detection (networks 1 and 2, respectively). (B) Examples of antennal grooming scores by the observer (absolute) and the system (probability); green and purple, top and bottom, respectively. Gray, 10 s 10 Hz red-light stimulus. (C) Mean absolute deviation of grooming detection between observer and system (mean ± SEM; % time of red-light stimuli; $n = 22$ animals). (D) Linear regression of grooming detection by system and observer (% time of red-light stimuli; probability ≥ 0.3; $n = 24$ animals). (E) Grooming detection quantified by observer and system (% time of red-light stimuli, $n = 24$ animals). Data, box (interquartile) and whisker (minimum and maximum, respectively) plot, with minimum, lower quartile (Q1), median, upper quartile (Q3), and maximum. Paired two-tailed t-student test. (F) Mean absolute deviation between manual and automatic grooming detection after axotomy (7 dpa; mean ± SEM; % time of red-light stimuli; $n = 22$ animals). (G) Linear regression of grooming detection between observer and system at 7 dpa (% time of red-light stimuli; probability ≥0.4; $n = 18$ animals). (H) Grooming detection quantified by observer and system at 7 dpa (% time of red-light stimuli, $n = 9$ animals). Paired two-tailed t-student test; ns, $p > 0.05$. Source data are available online for this figure.

tissue for TRAP, we identified *orco–Gal4*, which labels 800 olfactory receptor neurons with cell bodies housed in antennae and maxillary palps (Larsson et al, 2004). Their cell bodies can be readily ablated without killing the animals. However, we observed an increase in mortality in wild type but not in animals with preserved projections at 14 dpa (e.g., *orco⁺ dnmnat^OE*). To avoid the loss of animals, we used 7 dpa to perform TRAP and translatome analyses.

Our study identified around 500 enriched transcripts by TRAP in severed, preserved projections of adult flies at 7 days after axotomy. Many transcripts are associated with oxidative/reduction processes, a response known to be activated after axotomy to counteract the increased production of reactive oxygen species (ROS) (Llobet Rosell and Neukomm, 2019). Furthermore, we identified vesicle-mediated transport transcripts to facilitate pre- and postsynaptic communication. In addition, we isolated transcripts associated with RNA processing. RNA-binding proteins regulate mRNA axonal transport and local translation (Ederle and Dormann, 2017; Ishiguro et al, 2016; Thelen and Kye, 2020). This could suggest that translationally silent mRNAs may be stored locally and utilized to produce multiple copies of a protein when needed, providing an efficient response mechanism in emergencies (Jung et al, 2012). Among them was the RNA-binding protein 4F (Rnp4F), encoding an evolutionarily conserved RNA-binding protein. We also identified uncharacterized genes, *CG32533* and *CG1582*, which are predicted to possess RNA binding and helicase activity. Our findings thus add complexity to RNA processing and local translation.

In addition, our analysis revealed two types of transcriptional enrichments. We examined the transcripts in preserved projections compared to their wild-type controls and also considered the genetic background (e.g., wild type vs. *dnmnat^OE*). One GO-term class of transcripts, protein polyubiquitination, was enriched exclusively in severed preserved *dnmnat^OE* projections, as a marker for protein degradation. This increase suggests that protein degradation, generally in balance with protein synthesis, provides new amino acids for local translation (Ding et al, 2007; Jarome and Helmstetter, 2014). This enrichment indicates a response to axotomy, independent of the genetic background. In line with our observation, a recent study used *Sarm1^−/−* knockout axons to investigate mRNA decay, resulting in a comprehensive transcriptomic profile (Jung et al, 2023). The comparison of both datasets revealed a common significant enrichment of transcripts associated with protein ubiquitination. Thus, local translation could be activated as a response to axotomy in preserved projections. In contrast, all other GO-term classes were enriched in uninjured

*dnmnat^OE* controls, suggesting that *dnmnat^OE* changes the neuronal transcriptome.

Combining identified conserved mouse/*Drosophila* transcripts with a more stringent filtering approach revealed an enrichment of transcripts related to $Ca^{2+}$ transport/homeostasis. A conditional knockout (cKO) of *Nmnat2* in cortical glutamatergic neurons supports our observations, where transcriptomic analyses revealed that NMNAT2 loss results in a significantly reduced $Ca^{2+}$ transport/homeostasis (Niou et al, 2022). In line with our observations, elevated $Ca^{2+}$ levels can activate calpain proteases to execute axon degeneration (Yang et al, 2015). Intriguingly, voltage-dependent sodium channels are among the targets of calpains. The channel degradation leads to increased axonal $Ca^{2+}$ and subsequent axon degeneration (Iwata et al, 2004; Llobet Rosell and Neukomm, 2019; Rishal and Fainzilber, 2014).

Highly translated mRNAs are selectively degraded in axons; thus, their pools decrease over time (Jung et al, 2023). We used an RNAi-based screen to lower the neuronal mRNA pool. Before injury, in projections, the pool may offer sufficient mRNA substrates for local translation. However, 7 days post axotomy, the pool may be below the threshold where local translation is significantly reduced. There, it could result in reduced grooming behavior if the candidates are involved in sustaining synaptic function.

Interestingly, *traf4^RNAi* and *cac^RNAi* reduced grooming in non-axotomized animals. Since their projections appeared unaffected, it suggests that *traf4* and *cac* harbor a more general function in neuronal communication. In contrast, we identified candidates that resulted in reduced grooming solely after axotomy. Lowering their mRNA pools did not affect grooming in non-axotomized animals but resulted in reduced grooming specifically after axotomy. Since neuronal morphology remained unaffected, it is tempting to speculate that such candidates are locally translated to sustain synaptic function. Among them are genes in protein ubiquitination (*huwe1*, *CG10916*, and *CG6923*), and *nckx30c* involved in $Ca^{2+}$ homeostasis. Remarkably, we also observed increased grooming solely after axotomy in *calx^RNAi* animals. Further studies are required to dissect the underlying mechanism.

Identified candidates and their subsequent validation led us to speculate that local protein synthesis forms the basis for preserved synaptic function. Since we observed increased *raptor* transcripts in *dnmnat^OE* at 7 dpa, we conducted knockdown experiments targeting *raptor* and *s6k*, mediators of the mTORC1 pathway. The specific reduction of evoked antennal grooming behavior after axotomy with *s6k^RNAi* suggests an intricate involvement of the mTORC1 pathway in regulating preserved synaptic function through local translation.

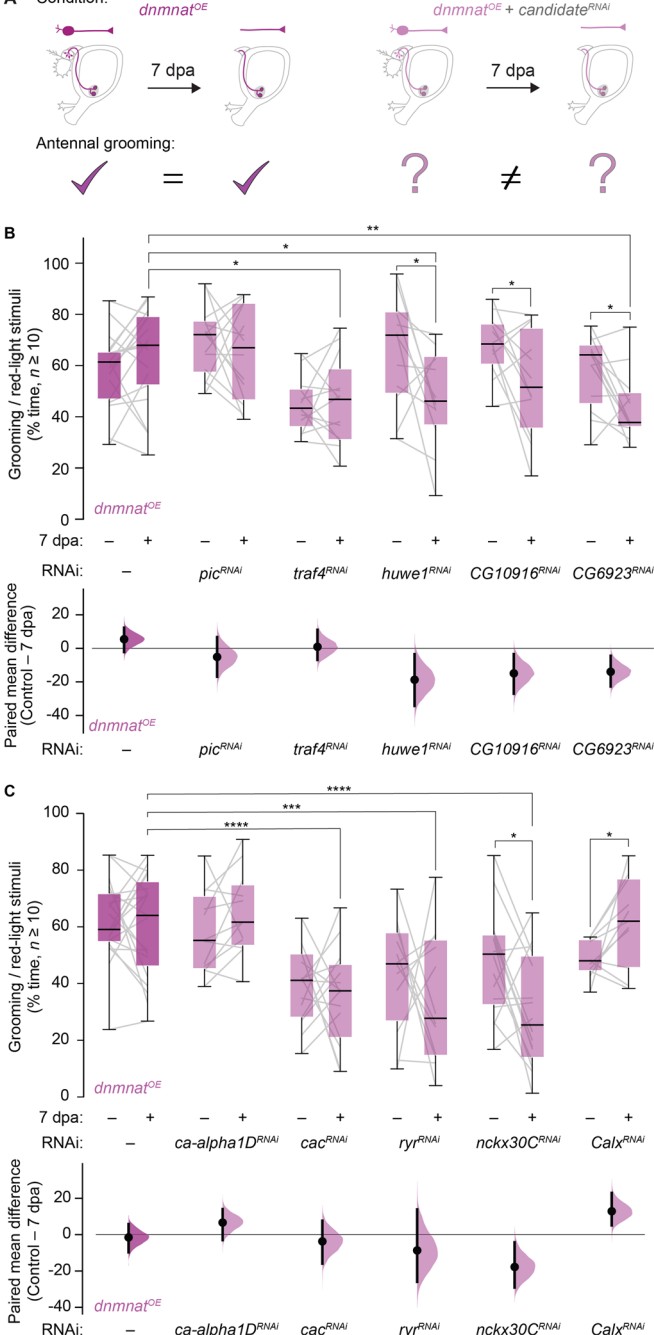

**Figure 5.** **RNAi-mediated knockdown of candidates involved in protein ubiquitination and Ca²⁺ transport required for preserved synaptic function 1 week after axotomy.**

functional adjustments of projections where a nucleus is far away or the soma cut-off. In some insects, the soma is entirely absent due to the selective evolutionary pressure of brain miniaturization, where anucleate axons continue to contribute to the behavior for the life span (Polilov, 2017, 2012). Interestingly, severed axons persist for weeks to months in various invertebrates and some vertebrates (Bittner, 1991, 1988). Here, local translation could ensure sustained maintenance of synaptic plasticity, which we observe in our model of impaired Wallerian degeneration.

Our comprehensive dataset revealed different uncharacterized genes in *Drosophila* linked to human diseases associated with impaired axonal transport and local translation. *CG13531* is an enigmatic gene with implications in axon extension and synaptic vesicle transport. Human ortholog(s) are predicted to be linked to Charcot-Marie-Tooth type 2X, amyotrophic lateral sclerosis type 5, and hereditary spastic paraplegia 11 (Yamaguchi et al, 2021; Yamaguchi and Takashima, 2018; Zhang et al, 2018). These observations highlight Wallerian degeneration, and the fruit fly, as powerful systems to gain further insights into human disease. Identifying and characterizing conserved fly/mammalian genes may therefore provide novel therapeutic avenues to slow or halt human axonopathies.

## Methods

### Reagents and tools table

| Reagent/Resource | Reference or Source | Identifier or Catalog Number |
|---|---|---|
| **Experimental Models** | | |
| *Drosophila melanogaster* | | RRID:NCBITaxon_7227 |
| *R60E02-Gal4* | Hampel et al, 2015 | |
| *orco-Gal4* | BDSC_26818 | |
| *or22a-Gal4* | BDSC_9952 | |
| *drp1-Gal4* | Neukomm et al, 2014 | |
| *ase−FLP²ᵇ* | Neukomm et al, 2014 | |
| *UAS-dnmnat* | BDSC_39699 | |
| *UAS-mCD8::GFP (II)* | Neukomm et al, 2014 | |
| *UAS-mCD8::GFP (III)* | Neukomm et al, 2014 | |
| *UAS-CsChrimson (X)* | BDSC_55134 | |
| *UAS-CsChrimson (II)* | BDSC_55135 | |

Based on our observations, we propose that severed projections with attenuated programmed axon degeneration employ local protein synthesis through mTOR signaling. Among the translated transcripts are candidates that help to cope with the turnover of already translated polypeptides by polyubiquitination and Ca²⁺ buffering. This model is supported by the impairment of either local translation or Ca²⁺ transport and polyubiquitination candidates, resulting in reduced synaptic function.

Altogether, our findings support the hypothesis of the "autonomous axon" (Alvarez, 2001), where local protein synthesis, with a pool of mRNAs, ensures the continued

| Reagent/Resource | Reference or Source | Identifier or Catalog Number |
|---|---|---|
| UAS–GFP::RpL10Ab | | BDSC_42683 |
| UAS–pic<sup>RNAi</sup> | | BDSC_33888 |
| UAS–traf4<sup>RNAi</sup> | | BDSC_55226 |
| UAS–huwe1<sup>RNAi</sup> | | BDSC_36715 |
| UAS–CG10916<sup>RNAi</sup> | | BDSC_36611 |
| UAS–CG6923<sup>RNAi</sup> | | BDSC_32448 |
| UAS–ca-alpha1D<sup>RNAi</sup> | | BDSC_33413 |
| UAS–cac<sup>RNAi</sup> | | BDSC_77174 |
| UAS–ryr<sup>RNAi</sup> | | BDSC_65885 |
| UAS–nckx30c<sup>RNAi</sup> | | BDSC_63570 |
| UAS–calx<sup>RNAi</sup> | | BDSC_28306 |
| UAS–raptor<sup>RNAi</sup> | | BDSC_41912 |
| UAS–s6k<sup>RNAi</sup> | | BDSC_42572 |
| **Recombinant DNA** | | |
| **Antibodies** | | |
| Magnetic agarose GFP-Trap beads | PROTEINTECH | GTMA-20 |
| Rabbit anti-GFP | Abcam | RRID:AB_305564 |
| Mouse anti-Tubulin | Sigma | RRID:AB_477593 |
| Chicken anti-GFP | Rockland | RRID:AB_1537402 |
| Mounse anti-Brp | DSHB | RRID:AB_2314866 |
| Goat anti-rabbit IgG (H + L) Dylight 800 | Thermo Fisher Scientific | RRID:AB_2633284 |
| Goat anti-mouse IgG (H + L) Dylight 680 | Thermo Fisher Scientific | RRID:AB_2633278 |
| Goat anti-chicken IgY (H&L) DyLight 488 | Abcam | RRID:AB_10681017 |
| Goat anti-mouse IgG (H + L) Alexa Fluor 546 | Thermo Fisher Scientific | RRID:AB_2737024 |
| **Oligonucleotides and other sequence-based reagents** | | |
| **Chemicals, Enzymes and other reagents** | | |
| Protease inhibitor | Sigma-Aldrich | P8340 |
| Nucleospin RNA plus kit | Takara Bio | U0984B |
| Nucleospin RNA plus kit XS | Takara Bio | U0990B |
| Cycloheximide | Nacalai Tesque | 06741-04 |
| Ribonucleoside vanadyl complexes | Sigma-Aldrich | R3380 |
| RNasin® Ribonuclease Inhibitor | PROMEGA | N2511 |
| KCl | Nacalai Tesque | 28538-75 |
| MgCl$_2$ | Nacalai Tesque | 20935-05 |
| NP-40 | Sigma-Aldrich | NP40S |
| DTT | Fujifilm Wako | 047-08973 |
| Tris | Thermo Fisher Scientific (Pierce) | 17926 |
| β-mercaptoethanol | Sigma-Aldrich | M6250 |
| Protein low-binding tube | Eppendorf | 0030 108.116 |
| All trans-retinal | Sigma-Merck | R2500-1G |
| **Software** | | |
| FastQC | http://www.bioinformatics.babraham.ac.uk/projects/fastqc | N/A |
| Trimmomatic | Bolger et al, 2014 | N/A |
| Hista2 | Kim et al, 2015 | N/A |

| Reagent/Resource | Reference or Source | Identifier or Catalog Number |
|---|---|---|
| SAMtools | Danecek et al, 2021 | N/A |
| HTSeq-count | Anders et al, 2015 | N/A |
| DESeq2 | Love et al, 2014 | N/A |
| DAVID | Huang et al, 2009 | N/A |
| R statistical software | https://www.R-project.org/ R v2022.07.2 + 576 | N/A |
| R package: dplyr | https://CRAN.R-project.org/package=dplyr | N/A |
| R package: pheatmap | https://CRAN.R-project.org/package=pheatmap | N/A |
| R package: ggplot2 | https://ggplot2.tidyverse.org | N/A |
| EthovisionXT17 (Noldus Information Technologies) | https://www.noldus.com/ethovision-xt | N/A |
| Python 3.7.3 (Python Software Foundation) | https://www.python.org | N/A |
| FIJI (ImageJ 1.54i) | https://doi.org/10.1038/nmeth.2019 | N/A |
| GraphPad Prism 10.2.3 | https://www.graphpad.com/ | N/A |
| Estimation Statistics | https://www.estimationstats.com/#/ | N/A |
| **Other** | | |
| Illumina NovaSeq 6000 SP Whole sequencer | Illumina | N/A |

## Husbandry and genetics

*Drosophila melanogaster* was used to perform experiments. Animals were raised on Nutri-Fly Bloomington Formulation food at 25 °C on a 12 h:12 h light-dark cycle unless otherwise specified; genotypes (Table EV1).

## Wing axotomy assay and axonal preservation analysis

### Wing assay
Animals were aged for 5–10 days at 20 or 25 °C. Wing injuries were applied as previously described (Llobet Rosell et al, 2022; Paglione et al, 2020). Briefly, one wing of each anesthetized animal was cut with micro scissors roughly in the middle, and was returned to an individual vial.

### Axonal preservation analysis
The distal, cut-off wing was mounted on a microscopy slide in Halocarbon Oil 27 and covered with a coverslip. The number of neuronal cell body clones in the cut-off wing was counted with an epifluorescence microscope. It represents the number of injured axons. The second wing was kept as uninjured control for subsequent analysis. To count intact or degenerated axons at 7 and 14 days post-axotomy (dpa), the injured and uninjured control wings were mounted and imaged using a spinning disk microscope.

## Antennal and maxillary palp ablation, brain dissection, immunohistochemistry, image acquisition and analysis

### Antennal or antennal and maxillary palp ablation
Animals were aged for 5–10 days at 25 °C. Anesthetized animals were subjected to bilateral antennal ablation or bilateral antennal

and maxillary palp ablation with high precision and ultra-fine tweezers (Paglione et al, 2020). Animals were recovered in food-containing vials for a specified time.

### Brain dissection
Adult brains were dissected at 7 and 14 dpa with the corresponding uninjured control, using a modification of a previously described protocol (Paglione et al, 2020).

### Immunohistochemistry
Decapitated adult heads were pre-fixed in 4% paraformaldehyde (PFA) in 0.1% Triton X-100 in 1x phosphate-buffered saline (PBS), pH 7.4 (PTX) for 20 min at room temperature (RT), and subsequently washed in PTX for 5 × 2 min. The dissection of the brains was performed in PTX, followed by brain fixation in 4% formaldehyde in PTX for 10 min at RT. After fixation, the brains were washed in PTX for 5 × 2 min and blocked in 10% normal goat serum (Jackson Immuno) in PTX (blocking solution) for 1 h at RT. Primary antibody incubation was performed in a blocking solution containing 1:1000 chicken anti-GFP (Rockland) and 2 µg/ml mouse anti-nc82 (DSHB) over night at 4 °C. Brains were washed with PTX for 3 × 10 min at RT and incubated with secondary antibodies in PTX, including 1:200 goat anti-chicken IgY (H&L) DyLight 488 (Abcam) and 1:500 goat anti-mouse IgG (H + L) Alexa Fluor 546 (ThermoFischer) for 2 h at RT. Brains were washed with PTX for 3 × 10 min at RT and mounted in Vectashield.

### Image acquisition and analysis
Images were acquired along the z-axis with a step size of 0.6 µm using a Nikon Spinning Disk confocal microscope. Image quantification was analyzed with ImageJ (NIH).

### Western blots
For each genotype, 20 adult heads were ground in 100 µl of Laemmli sample buffer (2 heads/10 µl) and heated for 10 min at 95 °C. 20 µl of the resulting lysate was loaded per well into 4–12% SurePAGE™ gels (GeneScript) with MOPS running buffer for higher molecular weight proteins. Protein separation was carried out at 80 V. Precision Plus Protein™ Kaleidoscope™ Prestained Protein Ladder (Bio-Rad) was used as a molecular weight marker. The proteins were wet-transferred onto PVDF membranes (Bio-Rad) at 110 V for 70 min at 4 °C. Membranes were washed with Tris-buffered saline (TBS) containing 0.1% Tween® 20 (Merk) (TBS-T) for 5 min at RT. Membranes were blocked with 5% milk (Carl-Roth) in TBS-T for 1 h at RT, and incubated in a blocking solution containing primary antibodies, including 1:5000 rabbit anti-GFP (Abcam), and 1:15000 mouse anti-Tubulin (Sigma), over night at 4 °C. Membranes were rinsed with TBS-T for 3 × 10 min and incubated with secondary antibodies in 5% milk in TBS-T, including 1:10,000 goat anti-rabbit IgG (H + L) Dylight 800, and 1:10,000 goat anti-mouse IgG (H + L) Dylight 680 (ThermoFisher), for 1 h at RT. Membranes were then rinsed with TBS-T for 3 × 10 min before signal acquisition. Fluorescent signals were acquired using Odissey® DLx (LI-COR), and image quantification was performed with ImageJ (NIH) through densitometric analysis.

### Translation ribosome affinity purification (TRAP)

#### Preparation of samples
The ribosomal pulldown was performed by adapting the already described protocol (Rozenbaum et al, 2018). In brief, for each genotype and condition, 300 adult animals were collected, snap-

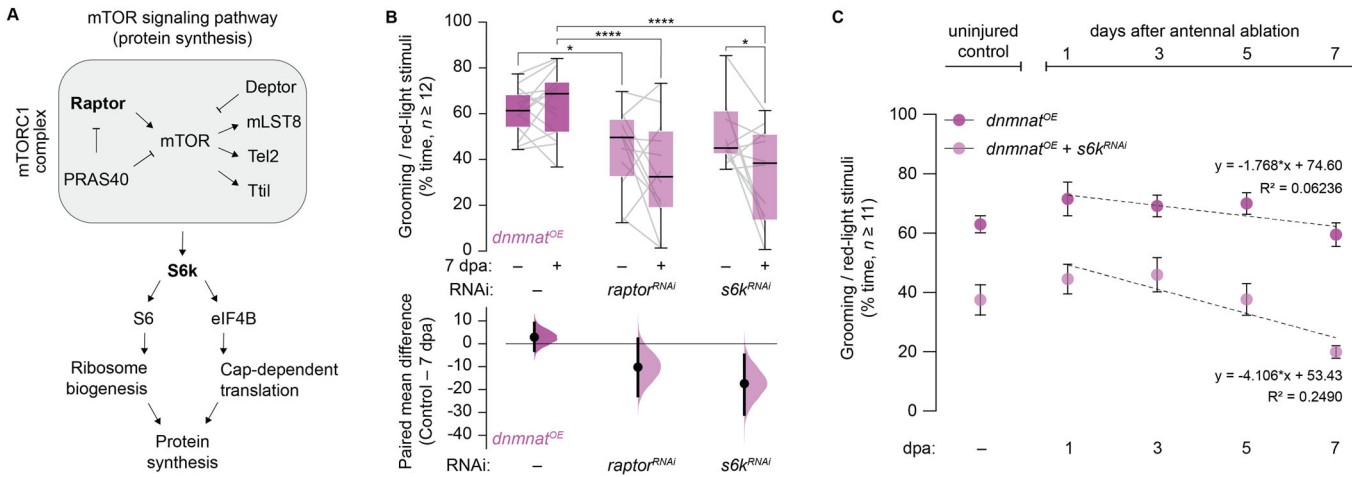

**Figure 6. RNAi-mediated candidate knockdown of mTOR signaling reduces preserved synaptic function 1 week after axotomy.**

(A) Protein synthesis mediated by the mTOR signaling pathway. Bold, candidates tested by RNAi. (B) RNAi-mediated knockdown of *raptor* and *s6k* result in reduced antennal grooming after axotomy. Top, Grooming as % time of red-light stimuli. Data, box (interquartile) and whisker (minimum and maximum, respectively) plot, with minimum, lower quartile (Q1), median, upper quartile (Q3), and maximum. Bottom, paired mean difference as bootstrap sampling distribution (dot, mean difference; vertical bar, 95% confidence interval, respectively, $n \geq 12$ animals). Data, % time of red-light stimuli ($n \geq 12$ animals). Paired two-tailed t-student test and two-way ANOVA with Dunnett's multiple comparisons test; ****$p < 0.0001$; *$p < 0.05$. (C) RNAi-mediated knockdown of *s6k* results in a faster reduction of antennal grooming after axotomy. Top, illustration of time-course analysis. Bottom, linear regression analysis of grooming (% time of red-light stimuli; $n \geq 11$ animals). Source data are available online for this figure.

frozen in liquid nitrogen and stored at −80 °C before processing. The frozen samples were homogenized using 2–3% weight per volume of homogenization buffer (100 mM KCl, 12 mM MgCl$_2$, 1% NP-40, 50 mM Tris, pH 7.4) supplemented with β-mercaptoethanol (10 μl/ml buffer solution). The lysate was centrifuged at 10,000 rpm for 10 min at 4 °C, and the supernatant was collected in a protein low-binding tube. 10% was kept for the input and stored at −80 °C.

### Preparation of beads

For each sample, 50 μl of magnetic agarose GFP-Trap beads (Proteintech, GTMA-20) were added to a RNase/DNase-free microcentrifuge tube. The beads were rinsed with homogenization buffer supplemented with 1 mM DTT, 150 μg/ml cycloheximide, 450 U/ml RNasin® Ribonuclease Inhibitor (Promega, N2511) 10 mM ribonucleoside vanadyl complexes (RVC, Sigma-Aldrich, R3380) and protease inhibitor (Sigma-Aldrich, P8340).

### GFP pulldown

The homogenized heads were incubated with magnetic agarose GFP-Trap beads (Proteintech, GTMA-20) to pulldown ribosomes with associated mRNAs at 4 °C over night. The beads were washed in high salt buffer (300 mM KCl, 12 mM MgCl$_2$, 1% NP-40, 1 mM DTT, 150 μg/ml Cycloheximide, 450 U/mL RNasin® Ribonuclease Inhibitor (Promega, N2511), protease inhibitor (Sigma-Aldrich, P8340), 50 mM Tris, pH 7.4) 3 × 5 min at 4 °C. Immediately after removing the final high salt wash buffer, samples were placed in a magnet, and 350 μl of lysis buffer was added to the beads (LB, RNA plus Kit for the inputs or LB1, RNA Plus XS kit, for the pulldowns). RNA was extracted from ribosomes bound to the beads according to the NucleoSpin RNA (Takara Bio, U0984B) (input) and RNA Plus XS (Takara Bio, U0990B) (pulldowns) kit manufacturer's recommendations.

### Library preparation, sequencing, and data analysis

The library preparation was performed with poly-A purification for input samples and skipped poly-A for the pulldowns, both without rRNA depletion. The libraries were sequenced on Pair End, using an Illumina NovaSeq 6000 SP Whole sequencer, with 150 bp determined. All reads were analyzed by a quality control check using FASTQC, followed by trimming with Trimmomatic (Bolger et al, 2014), where unpaired reads and adapter sequences were removed based on the quality scores. Then, another control quality check was performed.

Trimmed high-quality reads were aligned to the *Drosophila* genome (*Drosophila melanogaster*/dmel_r6.43_FB2021_06) using Hisat2 (Kim et al, 2015). The alignment files were sorted, indexed, and converted with Samtools to count reads with Htseq-Count (Anders et al, 2015). Library normalization and statistical analysis of differentially expressed genes (DEG) were analyzed by DESeq2 (Love et al, 2014), considering an adjusted *p*-value ≤ 0.05 and regularized logarithm (rLog) fold change smaller than −0.6 (downregulated) or greater than 0.6 (upregulated).

Gene Ontology (GO) term enrichment analysis for biological processes was performed with the Database for Annotation, Visualization, and Integrated Discovery (DAVID) 6.8 (Huang et al, 2009). Statistical analyses and data visualization were conducted in R using the base, dplyr, pheatmap, and ggplot2 packages.

In Figs. 2D and EV3, the following procedure was applied to each class: for example, under protein ubiquitination, significantly enriched classes were used to avoid eliminating potential

candidates: protein ubiquitination (GO:0016567) and protein ubiquitination involved in ubiquitin-dependent protein catabolic process (GO:0042787). The rLog data from the newly generated gene list was used to generate a heat map to assess transcript enrichment in different conditions (e.g., control and 7 dpa, respectively) and genotypes (e.g., wild type and *dnmnat$^{OE}$*, respectively). The analysis was repeated for each GO term class (Fig. 2C).

### Comparative analysis with published dataset

For the comparative analysis with the mammalian data, transcripts with adjusted *p*-value ≤ 0.05 and log2 Fold Change >0.6 (upregulated) were selected from Table S6B in (Jung et al, 2023). For rapid identification of orthologs, the list of the selected Ensembl Gene ID was analyzed with the *Drosophila* Integrative Ortholog Prediction Tool (DIOPT; http://www.flyrnai.org/diopt) (Hu et al, 2011). Only orthologs with high rank, best score, and DIOPT score ≥2 were chosen to compare with our dataset.

### Optogenetics-induced antennal grooming behavior

Genetic crosses were performed in the darkness on standard cornmeal food in aluminum-covered vials supplemented with 200 μM all-trans retinoic acid (Hampel et al, 2015; Paglione et al, 2020). Optogenetic experiments to induce antennal grooming were conducted in the dark. Animals were visualized and recorded using an 850 nm infrared light source at an intensity of 2 mW/cm$^2$ (Mightex, Toronto, CA). For CsChrimson activation, a 656 nm red-light source (Mightex) was used at 8 or 14 mW/cm$^2$ intensities. Animals were subjected to a 10 s 10 Hz red-light exposure, followed by a 30 s recovery phase where the red light was turned off. This repetition cycle was performed three times in total. Red light stimulus parameters were delivered with a NIDAQ board controlled through Bonsai (https://bonsai-rx.org/). Animals were manually scored using Noldus EthoVision XT software (Noldus Information Technology), where grooming activity was plotted as the percentage of time spent grooming the antennae during red-light stimuli. The ablation of the antennae did not damage the rest of the head or lead to the mortality of the animals. Animals that died during the analysis window (7–14 dpa) were excluded.

### Automated antennal grooming behavior detection system

Frame-wise scoring of antennal grooming behavior was performed as an image-recognition task. Briefly, bouts of antennal grooming are a stereotypical pattern of foreleg movements directed toward the antennae that span several consecutive frames. This pattern was captured and compressed on a single image by clipping the grooming episode over time and computing the pixel-wise standard deviation through the stack of frames. Supervised training of an artificial neural network was used to recognize these compressed images belonging to one of two categories: grooming or no-grooming. An end-to-end system was established that takes a raw video as input and returns a frame-wise probability as output. Two neural networks and a set of image-processing functions wrapped in a single system were established, requiring minimal user input. The details of the system,

together with the parameters used for training, validating, and testing are provided below.

## Network-1: detection of the head and its fore region

To simplify the task of detecting grooming events, the region of interest was restricted to the forehead of the animal and the region in front of it. A deconvolutional network was trained to detect the head of the animal and to output the coordinates of its centroid.

### Model
The deconvolutional network was developed in Python (V 3.7.4) using the PyTorch Library (V 1.13.0). The network is composed of six layers: 3 convolutional layers (channels: 32, 64, 128; kernel size: 3, 5, 3; max pooling: 2; stride: 1) and 6 deconvolutional layers (channels: 128, 64, 32; kernel size: 3, 5, 3; max unpooling: 2; stride: 1). We used the Kaiming weight initialization and ReLU activation functions as implemented in the Pytorch library.

### Dataset
Raw videos are large ($1280 \times 1024$ px) grayscale avi files. To reduce processing time, the network was trained, validated, and tested on lower resolution frames ($320 \times 256$ px) extracted from the video files. 916 tuples (raw frame, binary mask of the head position) were used for training, 229 tuples for validation, and 250 tuples for testing the accuracy of the network.

### Training, validation, and test
The network was trained to minimize the Mean Squared Error loss function with the ADAM optimizer on batches of 8 tuples for 14 epochs. The learning rate was fixed at 0.003. The coordinates of the binary mask's centroid was used to probe the accuracy on the validation set. The Euclidean distance between the centroid of the binary mask and the max value of the confidence map produced by the network is used to evaluate the accuracy of the segmentation process. The distance (average ± standard deviation) between centroids is $3.88 \pm 3.48$ pixels during validation and $4.33 \pm 10.43$ pixels in the test set. The minimum diameter of an animal's head (e.g., side view relative to the camera) was estimated ~32 pixels. For this reason, a cut-off radius of 8 pixels (e.g., max distance between ground truth centroid and predicted centroid) was used to define the segmentation accuracy. According to this cut-off, the network correctly predicted the position of the head in 98.8% of the test frames. The text file output of this network contains the frame-by-frame coordinates of the predicted position of the centroid of the head. In addition, the inference script automatically detects the onset/offset of the red-light stimuli by first taking the average pixel intensity of a $100 \times 50$ pixel region surrounding the LED on the initial 200 frames (light-off) and then computing the z-score for each frame in the video. Frames that are 2 z-scores above the average are defined as light-on frames. This procedure leads to a marginal underestimation (5.4 frames, mean absolute difference) of the total number of light-on frames as compared to the measure taken by a human observer.

## Network-2: grooming detection

The output of the deconvolutional network (e.g., the coordinates of the head's centroid) was used to define an ROI centered on the head of the animal and its fore region. Following this crop, the

pixel-wise standard deviation of 3 consecutive frames was computed, and the network classified the resulting image as either grooming or no-grooming. A sliding window (stride = 1 frame) was applied to the input to compute the frame-wise probability of grooming on the full length of the raw video. The network was initially trained on naive animals. However, the inference on animals that underwent axotomy (e.g., antennal ablation) was suboptimal. Therefore, the network was trained on two distinct datasets (e.g., animals with or without antennae), leading to two distinct sets of weights that were independently deployed according to the experimental group (e.g., before or after axotomy). The animal condition did not affect the quality of the inference on head tracking performed by the deconvolutional network (Network-1).

### Model
Pretrained ResNet50 (Pytorch: ResNet50_Weights.IMAGEN-ET1K_V2) fine-tuned on our training datasets.

### Dataset
Antennal grooming is defined as a stereotyped circular movement of the forelegs sweeping over the surface of the antennae. A trained experimenter manually labeled the grooming episodes using the manual scoring module of EthovisionXT (Noldus Information Technologies). This dataset represented the ground truth for training the ResNet50 and it required further processing before being used to train the network. First, the raw videos were processed with the deconvolutional network (see section "Network-1: detection of the head and its fore region") to extract the coordinates of the head's centroid. Manual scoring of grooming events together with the coordinates of the head centroid were used to clip grooming bouts and crop frames ($128 \times 128$ px) on a region centered on the head of the animal. For each grooming bout, the pixel-wise standard deviation was computed on a stack of 3 consecutive frames taken with a sliding window (stride = 1) over the bout. A sequence of 3 frames was used to capture short bouts (camera fps: 15 Hz, ~0.7 s) of grooming. The same process was repeated for frames labeled as "no_grooming". These sets of images (e.g., grooming and no_grooming) were further inspected to remove images that could not be unambiguously attributed to one of the two categories. The images were then randomly allocated to one of three sets: train, validation, or test. Two distinct sets of images were obtained from the before- and after-axotomy conditions. The before-axotomy set comprised 30 subjects (number of frames, train = 3499, validation = 943, test = 64), while the after-axotomy comprised 22 subjects (train = 1615, validation = 413, test = 495).

### Training, validation, and test
The ResNet50 was trained for 41 epochs with a Cross Entropy loss function, the ADAM optimizer, and a decaying learning rate (gamma = 0.1, 7 epochs). The classification accuracy on the validation and test sets was 95.9% and 93.5%, respectively.

## Post-processing of grooming prediction

The ResNet50 model outputs a text file containing the frame-wise probability values for grooming behavior. However, because of the projection through time (3 consecutive frames) of the training set and the sliding window ($n = 1$), the probability value failed to capture sharp transitions between grooming states (e.g., onset-offset). For this reason, first, inference on the training set videos

was run to get raw probability values; then these were binarized according to 7 probability threshold values $\theta$ ranging from 0.1 to 0.9 (Fig. 4C,F) and for each frame with $p > \theta$, the 3 consecutive frames were labeled as 'grooming'. This procedure allowed to compensate for the bias introduced by the compression of 3 consecutive images without producing excess false positives. Finally, comparing the resulting grooming scores with manual scores allowed us to find an optimal probability threshold value that minimizes the difference between system and observer scores.

## Replication

Unless stated otherwise, at least 3 biological replications were performed for all experiments for each genotype and/or condition.

## Software and statistics

Image-J and Photoshop were used to process wing, $or22a^+$, $JO^+$, and $orco^+$ images. The optogenetics section includes the software used for the analysis. The comparison between various observers vs. system, as well as the paired mean difference of automated antennal grooming detection, were plotted using https://www.estimationstats.com/#/. Unless otherwise stated, GraphPad Prism 10.2.3 was used for statistical analyses.

## Data availability

The code and scripts for running the automated grooming detection system from this publication are available on the GitHub database: https://github.com/Neukomm-lab/automated-antennal-grooming. The TRAPseq data is available on the Gene Expression Omnibus database with the assigned identifier GSE270011: https://www.ncbi.nlm.nih.gov/geo/query/acc.cgi?acc=GSE270011.

The source data of this paper are collected in the following database record: biostudies:S-SCDT-10_1038-S44319-024-00301-8.

## Peer review information

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

## Acknowledgements

We thank Dr. Takakazu Yokokura and Dr. Luigi Bozzo for the access to and support in the OIST fly facility and the Cellular Imaging Facility (CIF, FBM, UNIL). This work was supported by the following grants and fellowships: two Swiss National Science Foundation (SNSF) Assistant Professor starting grants (PP00P3_176855 and PP00P3_211015), the International Foundation for Research in Paraplegia grant (P180), and the SNSF Spark grant (190919) to LJN; the Japan Society for the Promotion of Science (JSPS) fellowship (GR20107) to MP, and the Japan Society for the Promotion of Science grant (23K27107) to MT.

## Author contributions

**Maria Paglione**: Conceptualization; Data curation; Formal analysis; Validation; Investigation; Visualization; Methodology; Writing—original draft; Project administration; Writing—review and editing. **Leonardo Restivo**: Software; Supervision; Methodology; Writing—review and editing. **Sarah Zakhia**: Software; Formal analysis; Methodology; Writing—review and editing. **Arnau Llobet Rosell**: Investigation; Writing—review and editing. **Marco Terenzio**: Resources; Formal analysis; Supervision; Investigation; Methodology; Writing—review and editing. **Lukas J Neukomm**: Conceptualization; Resources; Supervision; Funding acquisition; Writing—original draft; Project administration; Writing—review and editing.

Source data underlying figure panels in this paper may have individual authorship assigned. Where available, figure panel/source data authorship is listed in the following database record: biostudies:S-SCDT-10_1038-S44319-024-00301-8.

## Disclosure and competing interests statement

The authors declare no competing interests.

# Expanded View Figures

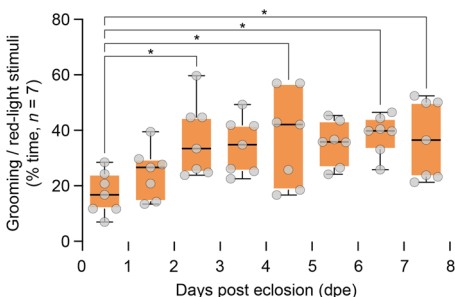

**Figure EV1. Time-course of optogenetically-evoked antennal grooming behavior in wild type after eclosion.**

Quantification of manually scored antennal grooming in wild-type flies with CsChrimson expressed in JO neurons between 0 to 7 days post eclosion (dpe). Data (% time of red-light stimuli; $n = 7$ animals), box (interquartile) and whisker (minimum and maximum, respectively) plot, with minimum, lower quartile (Q1), median, upper quartile (Q3), and maximum. One-way ANOVA with Tukey's multiple comparisons test. *$p < 0.05$.

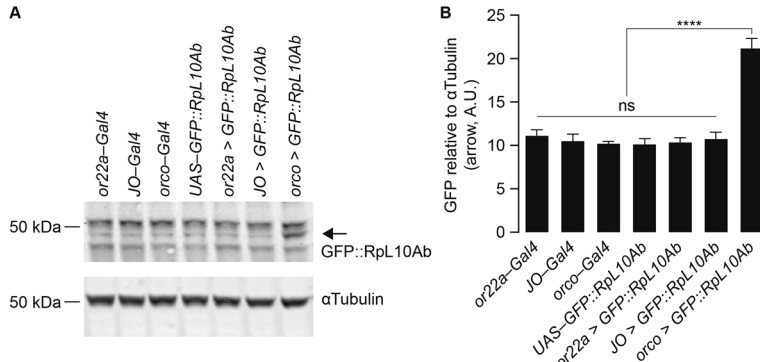

**Figure EV2. Olfactory organ projections as a large-scale source to express and detect GFP-tagged ribosomal protein 10Ab.**

(A) Western blot of *Drosophila* heads with GFP-tagged ribosomal protein 10Ab (GFP::RpL10Ab) expressed in *orco*⁺ neurons. 4 heads/lane; arrow, molecular weight of GFP::RpL10Ab. (B) Quantification of GFP immunoreactivity by densitometry. Data, mean ± SEM ($n = 3$, three replicates of three biological experiments). Arbitrary units, A.U.; One-way ANOVA with Tukey's multiple comparisons test; ****$p < 0.0001$; ns (not significant), $p > 0.05$.

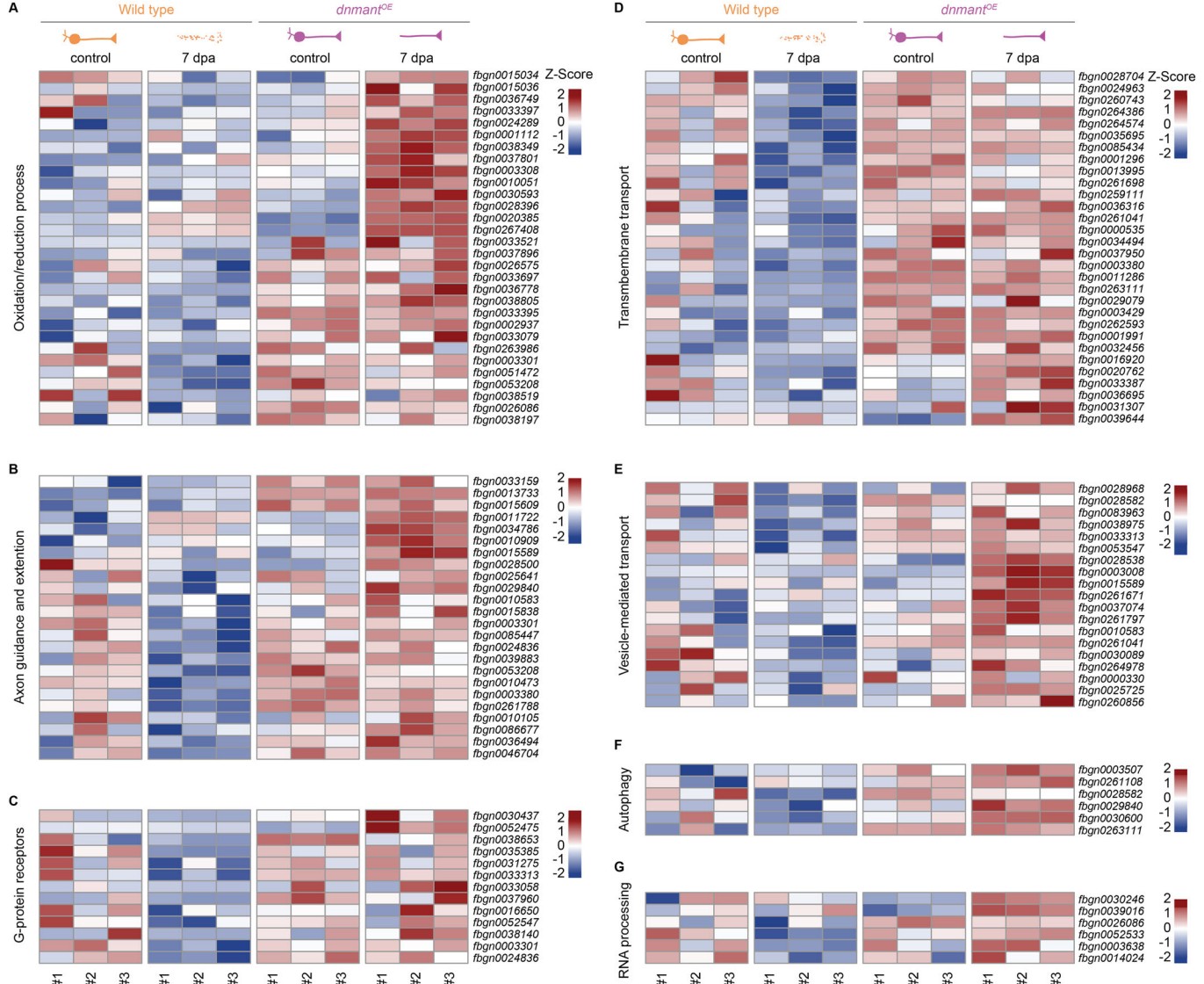

**Figure EV3. Heat maps of biological process GO terms significantly enriched in *dnmnat^OE* 7 dpa.**

(A) Heat map of oxidation/reduction process transcripts ($n = 30$). (B) Heat map of axon guidance and extension transcripts ($n = 24$). (C) Heat map of G-protein receptor transcripts ($n = 13$). (D) Heat map of transmembrane transport transcripts ($n = 30$). (E) Heat map of vesicle-mediated transport transcripts ($n = 19$). (F) Heat map of autophagy transcripts ($n = 6$). (G) Heat map of RNA processing transcripts ($n = 6$). Expression levels, Z-Score average.

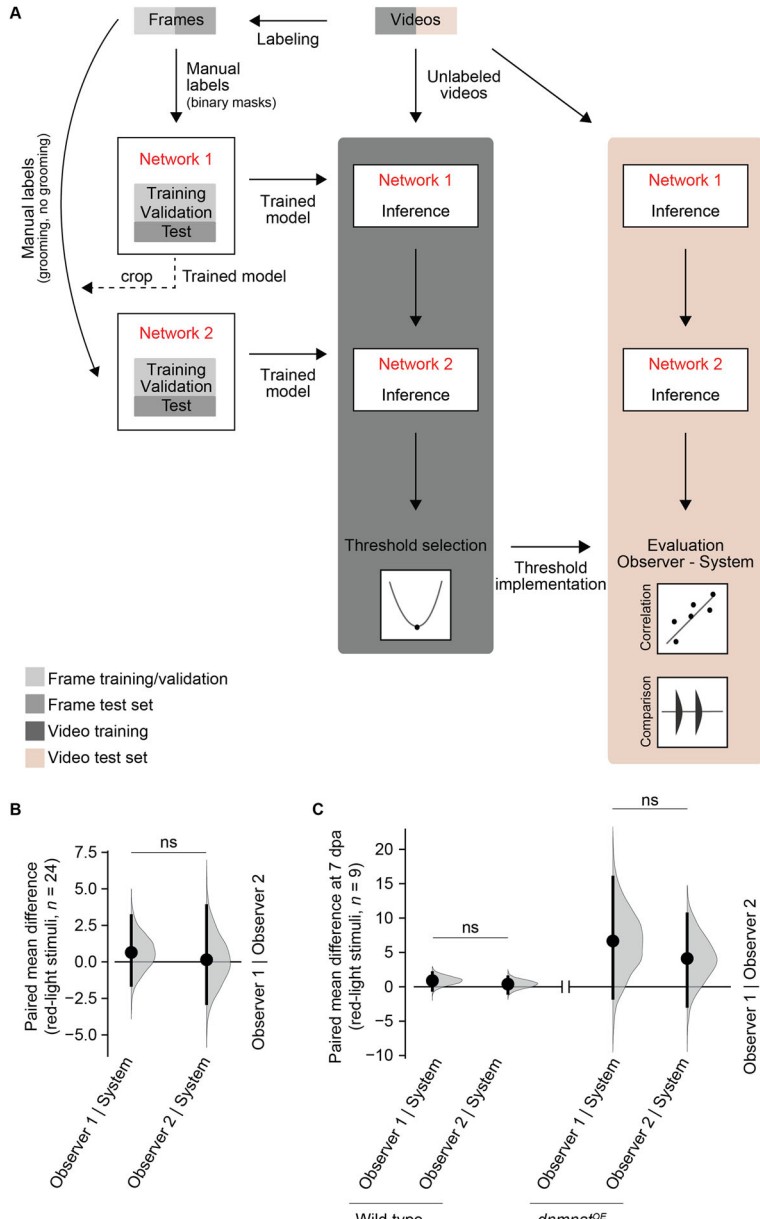

**Figure EV4. Dual-network training and evaluation pipeline for automated grooming detection and comparison with observer scores.**

(**A**) Illustration of Network 1 and 2 training and testing pipelines. Raw videos were split into training and test sets (gray and red, respectively). Video frames from the training set were manually labeled to identify the position of the head in the image (e.g., binary mask) and assign the grooming or no grooming labels. Network 1 was trained, validated, and tested on its own training set (frames). Network 2 was trained, validated, and tested on the labeled frames cropped by Network 1. Both networks, once trained, were used to determine the optimal threshold value of grooming detection using the same training set (raw videos). Finally, the test set (raw videos) was used to evaluate the accuracy of the system by comparing its performance with the scores of the observer. (**B**) Standard deviation between grooming scores given by each observer to the same uninjured wild-type animal ($n = 24$ animals). (**C**) Standard deviation between grooming scores given by each observer to the same wild type and *dnmnat^{OE}* at 7 dpa ($n = 9$ animals). The line break denotes the separation between the controls for each genotype. The plots illustrate the similarity between the standard deviations of observer 1 versus the system and observer 2 versus the system, as compared to the standard deviation observed among observers (horizontal black line, mean, number of frames). Non-parametric one-way ANOVA (Friedman test) with Dunn multiple comparisons test; ns, $p > 0.05$.

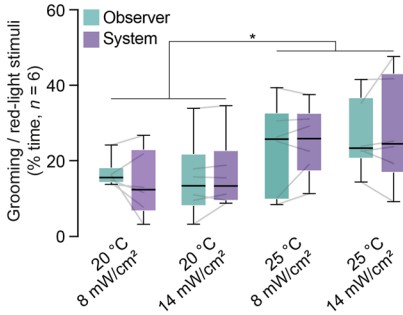

**Figure EV5. Detection of temperature-dependent changes in antennal grooming.**

Quantification of antennal grooming detection by observer and system in wild-type flies with CsChrimson expressed in JO neurons of 7-day-old animals (green and purple, respectively; probability, ≥0.3; % time of red-light stimuli; $n = 6$ animals). Data, box (interquartile) and whisker (minimum and maximum, respectively) plot, with minimum, lower quartile (Q1), median, upper quartile (Q3), and maximum. Three-way ANOVA; *$p < 0.05$.

