## [Peer Review File · EMBO Reports]

Local translome sustains synaptic function in impaired Wallerian degeneration

Maria Paglione, Leonardo Restivo, Sarah Zakhia, Arnau Llobet Rosell, Marco Terenzio, and Lukas Neukomm

Corresponding author(s): Lukas Neukomm (lukas.neukomm@unil.ch)

Review Timeline:

Submission Date:	10th Feb 24
Editorial Decision:	8th Apr 24
Revision Received:	5th Aug 24
Editorial Decision:	19th Sep 24
Revision Received:	7th Oct 24
Accepted:	17th Oct 24

Editor: *Esther Schnapp*

Transaction Report:

Dear Prof. Neukomm,

Thank you for your patience while your ms was peer-reviewed at EMBO reports. We have now received the full set of referee reports that is pasted below.

As you will see, all referees acknowledge that the findings are interesting. They also have a few suggestions for how the study can be further improved, and I think that all should be addressed. Please let me know in case you disagree, and we can discuss the exact revision requirements further, also in a video chat, if you like.

I would thus like to invite you to revise your manuscript with the understanding that the referee concerns must be fully addressed and their suggestions taken on board. Please address all referee concerns in a complete point-by-point response. Acceptance of the manuscript will depend on a positive outcome of a second round of review. It is EMBO reports policy to allow a single round of major revision only and acceptance or rejection of the manuscript will therefore depend on the completeness of your responses included in the next, final version of the manuscript.

We realize that it is difficult to revise to a specific deadline. In the interest of protecting the conceptual advance provided by the work, we recommend a revision within 3 months (9th Jul 2024). Please discuss the revision progress ahead of this time with the editor if you require more time to complete the revisions.

You can either publish the study as a short report or as a full article. For short reports, the revised manuscript should not exceed 29,000 characters (including spaces but excluding materials & methods and references) and 5 main plus 5 expanded view figures. The results and discussion sections must further be combined, which will help to shorten the manuscript text by eliminating some redundancy that is inevitable when discussing the same experiments twice. For a normal article there are no length limitations, but it should have more than 5 main figures and the results and discussion sections must be separate. In both cases, the entire materials and methods must be included in the main manuscript file.

- 1) A data availability section providing access to data deposited in public databases is missing. If you have not deposited any data, please add a sentence to the data availability section that explains that.
- 2) Your manuscript contains statistics and error bars based on $n=2$. Please use scatter blots in these cases. No statistics should be calculated if $n=2$.

5) a complete author checklist, which you can download from our author guidelines <<https://www.embopress.org/page/journal/14693178/authorguide>>. Please insert information in the checklist that is also reflected in the manuscript. The completed author checklist will also be part of the RPF.

6) Please note that all corresponding authors are required to supply an ORCID ID for their name upon submission of a revised manuscript (<<https://orcid.org/>>). Please find instructions on how to link your ORCID ID to your account in our manuscript tracking system in our Author guidelines <<https://www.embopress.org/page/journal/14693178/authorguide#authorshipguidelines>>

7) Before submitting your revision, primary datasets produced in this study need to be deposited in an appropriate public database (see <https://www.embopress.org/page/journal/14693178/authorguide#datadeposition>). Please remember to provide a reviewer password if the datasets are not yet public. The accession numbers and database should be listed in a formal "Data Availability" section placed after Materials & Method (see also <https://www.embopress.org/page/journal/14693178/authorguide#datadeposition>). Please note that the Data Availability Section is restricted to new primary data that are part of this study. * Note - All links should resolve to a page where the data can be accessed. *
If your study has not produced novel datasets, please mention this fact in the Data Availability Section.

I look forward to seeing a revised form of your manuscript when it is ready.

Yours sincerely,

Referee #1:

Review of Paglione et al, 2024 EMBO Report.

In this elegant study, Paglione et al study synaptic conservation upon inhibition of Wallerian Degeneration (WD) following severing of the soma. They use *Drosophila* as an excellent genetic model system which is absolutely appropriate. First, they establish the protection ability of overexpression of dNMNAT in severed neurons. Their results are somewhat different from a previous study from the Freeman lab, but they propose that the stronger WD protection that they observe is likely to the fact that they used an untagged dNMNAT transgene (in contrast to myc-tagged), as before, or because of higher expression. While they never proved this, I don't think it's important to address. Then they also developed a new model for WD, of the JO+ neurons - this is an excellent choice as it allowed them to assay for synaptic conservation (function of the circuit - i.e. antennal grooming). Next, they decided to perform an analysis of the translating transcriptome, using the TRAP technique. In short - they expressed GFP tagged Ribosome subunits and then isolated bound transcripts from severed axons. This allowed them to identify genes that are likely translated in axons and upregulated following injury. In figure 3, the authors develop a very cool automatic screen to measure optogenetic-induced grooming before and after injury and upon KD of candidate genes. Figures 4-5 focus on the results from that screen that identify genes involved in ubiquitination, calcium homeostasis, as well as TOR, as required for optogenetic-induced grooming following injury. I think the study demonstrated an impressive tour-de-force and on route generation of new tools and methods. As such, I support publication in EMBO reports pending the clarifications/corrections of my concerns as detailed below.

By far the weakest figures are #4+#5. The fact that the transcript were identified in a TRAP experiments using severed axons (i.e. translating transcripts are present in the axons), and that their KD using RNAi (throughout the cell, including the soma, and throughout development including pre- and post-injury) does NOT prove, in my opinion, that their requirement for synaptic maintenance DEPENDS on their translation in axons. Unless I am missing something - there is no easy way around this experimentally. Thus, I would opt to change the text to more careful wording and use the discussion to promote the hypothesis that is spelled out (prematurely, in my opinion) at the very end of the results.

Second - and perhaps most importantly - one of the most important new claims here (apart from the tools/methods developed - which merit publication by themselves), is that axon preservation and synaptic preservation are not the same thing. i.e. they claim that while local translation is not required for NMNAT inhibition of axonal WD (as was previously shown - but not demonstrated in their own hands) but is required for NMNAT inhibition of synaptic elimination during WD. There is a crucial, yet missing, experiment that needs to prove this. They need to show that while the antennal grooming is NOT preserved upon KD of selected genes (most notably Nckx30C, Raptor and S6K), the degeneration of the axonal projections IS inhibited.

Apart from that - the paper is well written and compelling. Some minor textual/presentation suggestions (and completely up to the authors/editor): 1) in the introduction it sounds as if WD is regulated only by 4 genes - this is the current state and adding the words 'so far' would make it clearer. 2) two schemes, describing the wing injury experiment (otherwise, 1cb is quite cryptic to the reader who is unfamiliar with the system), and the grooming experiment (1G) - would help. 3) the very last sentence in the results would benefit from more accurate punctuation (I also suggest to move it to the discussion, as discussed above) - at the moment it is written as a causal chain of events, but there is no proof for that.

Referee #2:

The NAD-dependent axon destruction program is a relatively newly discovered and evolutionarily conserved biological pathway that is of broad interest not only to the community of molecular biology and neurobiology but also to clinical scientists, because its prevention (for example, by Sarm1 inhibition or Nmnat2 over-expression) may delay neurodegenerative diseases.

In this manuscript, Paglione and colleagues report the possibility of locally synthesized proteins, such as huwei, CG10916, nckx30C, and s6k, support synaptic function in severed, Wallerian degeneration (WD)-delayed axons in *Drosophila*. They generated UAS-GAL4 fly lines, in which dnmnat is over-expressed in GAL4-expressing neuronal populations. They showed that dnmnat over-expression prevents WD at least for two weeks in four different GAL4 lines and that optogenetic activation of WD-delayed, severed axons elicits antennal grooming behavior, a proxy of intact synaptic transmission, in one line (JO+ neurons), which they detect using the automated image analysis method they developed. For biochemical purification of ribosome-bound mRNAs from severed axons, they used the orco-GAL4 line that labels more neurons and axons and found that some mRNAs

are enriched in axons after axotomy. RNAi-mediated depletion of some of those mRNAs in JO+ neurons prevented the preservation of antennal grooming behavior after axotomy, suggesting that local translation of those mRNAs may contribute the sustained synaptic transmission after axotomy.

This is a well-performed study reporting several interesting findings, tools, and datasets that may be of interest to the community of molecular biology and neurobiology. The manuscript is written in an engaging way and the discussion is comprehensive. I have the following comments that the authors may consider in making this study more accessible to readers.

<Major comments>

1. The requirement of local translation in synaptic function

Blocking axon destruction program delays WD, and after such manipulations, severed axon terminals retain intact morphology and function. How synaptic function is maintained over weeks without the supply of new materials from the soma is a key question in the field. The most important finding of this manuscript, in this sense, is perhaps that proteins synthesized in severed and WD-delayed axons support synaptic function over weeks.

The authors used TRAP in the first part of the manuscript to identify mRNAs whose axonal translation increases after injury. In the latter part of the manuscript, the authors used RNAi-mediated knockdown of the candidate mRNAs to investigate the requirement of these genes in synaptic function. The authors interpreted these results as some of the mRNAs on which they performed functional analyses support synaptic function by local translation.

Although this interpretation is logical, the data do not directly show that local translation of these mRNAs is required for sustained synaptic transmission, because RNAi was performed before axotomy and in the entire neuron including the soma.

While the study suggests that locally synthesized proteins support synaptic function in severed axons, the interpretation of the RNAi-mediated knockdown experiments should be cautious. Conducting knockdown experiments specifically in axons or after axotomy in *dnm1atOE* neurons would strengthen the evidence for the requirement of local translation. If this is not feasible, the authors should tone down the statement such as the last sentence of page 10 ("Thus, our RNA-mediated approach provides evidence that the local translation of ...") and qualify the interpretation of these experiments.

2. Possible new experiments

Because the main message of this manuscript is the local translation in pre-synaptic function, I think that alternative functional experiments could be performed. For example, the authors could perform time-course experiments on antennal grooming behaviors shortly after axotomy.

Repeated (or continuous) optogenetic stimulation on the day of axotomy will lead to depletion of pre-synaptic materials and premature cessation of stimulation-induced antennal grooming behavior (perhaps on the same day of axotomy). Recovery of the behavior will indicate replenishment of presynaptic materials, and if it happens by local translation, it may be possible to block it either by cycloheximide or RNAi-mediated depletion of the effective genes (e.g., *huwei*, *cg10916*, *nckx30C*, and *s6k*).

3. The results of TRAP experiments

The results of DEG experiments should be provided as supplemental tables, and whenever possible the gene names or symbols should be annotated in figures (e.g., Figure S3).

<Minor comments>

4. Figure 1A

- 1) Both upper and lower panels are labeled with "1 cb", but it is difficult to see where the cell bodies are located in the figure.
- 2) In the wild type, one axon appears intact even 14 days after axotomy. Are some of the *Dpr1+* neurons resistant to Wallerian degeneration?

5. Figure 2A

It is difficult to appreciate where the cell bodies and axon terminals are located in the diagrams of the head. I understand that the large grey circle inside the head indicates the antennal lobe, which does not contain the cell bodies of *orco+* neurons. But the small green circles may be mistaken for the cell bodies. The authors should annotate the diagrams to clarify the head contains no cell bodies of the labeled neurons.

6. Background group in TRAP

The authors used the GAL4-only group as the background control. It is not specified whether axotomy was performed and whether TRAP was performed (if so, when TRAP was performed in relation to axotomy).

7. Polyribosomes

Unless the authors have evidence that what they pulled down by TRAP are polyribosomes, the term should be replaced with ribosomes.

8. Discussion

The Discussion section is interesting but has paragraphs that are not directly relevant to the findings of the manuscript (such as those on Rabies viruses and hibernating vertebrates). On the contrary, the discussion on the genes whose RNAi-mediated knockdown leads to defective development/assembly of synapses (such as *traf* and *cac*) or the maintenance of synaptic function after axotomy (such as *huwei* and *s6k*) is missing. The authors should expand the discussion on points that are relevant to their findings.

9. Incomplete sentence

The sentence starting with "In JO+ *dnmnat*OE neurons, RNAi-mediated" on page 11 appears to be incomplete.

10. Citation

"ZenXian Nious, 2002" is formatted in a different way than the other references.

Referee #3:

Evidence, reproducibility and clarity:

In the present study, Paglione et al. describe the effects of local translation on programmed axon degeneration (Wallerian degeneration) in a *Drosophila* model of axotomy. They utilise a *dnmnat* overexpression system to preserve injured axons and identify transcripts that differentially interact with ribosomes when the NAD⁺ homeostasis is perturbed. Here, they find transcripts encoding proteins with key roles in mTORC1 pathway, protein ubiquitination and calcium homeostasis, a similar set of biological processes dysregulated in mammalian models of Wallerian degeneration. They also present a novel high-throughput behaviour assay tool to assess synaptic activity in the JO+ circuit, which was validated with an impressive set of control experiments. Knockdowns of preferentially translated mRNAs identified from the *dnmnat* OE system lead to reduced level of induced grooming behaviour, consistent with the reversal of programmed axon degeneration to the 'normal' rate. Overall, the key conclusions of the paper align with the conducted experiments, but a few additional experiments or clarification of the methods may be necessary to provide more robust conclusions.

Significance:

This work offers novel insights into the process of axon preservation although its molecular mechanism of action is not well characterised. The authors identify a group of locally translated and evolutionary-conserved transcripts that respond to altered NMD/NAD⁺ ratio, which has been shown to trigger Wallerian degeneration. Together with the deep learning-aided grooming behaviour paradigm, this work will be valuable in dissecting the precise mechanism of pathological axon degeneration/preservation.

Major comments:

1) In Figure 2D, I don't think the experiment supports the claim "We identified transcripts of protein ubiquitination enriched in *dnmnat*OE solely after injury". Here the authors are comparing translation profiles of <entire neuron> vs <neurites>. This can be a problem because the translation level in cell bodies could overwhelm the level in neurites, as the authors mentioned in Page 6. A proper comparison should be set in the same sub-compartment e.g. <non-ablated condition neurites> vs <ablated condition neurites>, or should be limited to the same compartments. Comparison of whole-neuron transcriptome between wildtype and *dnmnat*OE, however, is very useful.

2) In Page 8, the additional background filtering method should be better clarified. Is the background control library (*orco*-Gal4) an RNA-seq library? Was the entire adult head used for the library preparation? Comparing RNA-seq and TRAP-seq libraries often requires a different normalisation approach, which should be described in the Methods. If the aim was to filter out transcripts with low/non-reproducible readcounts, why not use a simple transcripts per million cut-off from TRAP-seq?

3) In Figure 4, can the authors show images of accelerated axon degeneration (as in Figure 1) in conditions where grooming behaviours are affected? The morphological data will nicely complement the behavioural information.

Minor comments:

1) Tables of TRAP-seq DESeq2 output and DAVID gene ontology analysis should be included as supplementary files.

2) TRAP-seq dataset should be uploaded to public repositories (e.g. Gene Expression Omnibus).

3) In Figure S3, it would be helpful to show the name of the genes next to the heatmaps since only a handful of genes are displayed per GO category.

4) TRAP-seq was performed in the *Orco*⁺ system while the behavioural assay was done in JO⁺ neurons. Can the authors comment/discuss conservation of axon degeneration pathways in neuron subtypes? Are the genes with no RNAi behavioural phenotype expressed in both systems?

5) In Figure 4, are the RNAi lines used for the knockdown experiments validated?

6) In the Method section regarding mouse/fly orthologs, how was the high rank assessed? Did the authors use a particular DIOPT score threshold?

EMBOR-2024-58968-T

Rebuttal letter

We would like to thank all three referees for their positive and insightful comments, and fair criticism. Their feedback helped us to substantially improve the quality of our manuscript. Below, we provide a point-by-point response to each major and minor comment. Edits/changes are indicated in red in the revised, resubmitted manuscript, and excerpts are outlined below each comment. We also note that we changed injury to axotomy and calcium to Ca^{2+} throughout the manuscript.

Referee #1:

Review of Paglione *et al*, 2024 EMBO Report.

In this elegant study, Paglione *et al* study synaptic conservation upon inhibition of Wallerian Degeneration (WD) following severing of the soma. They use *Drosophila* as an excellent genetic model system which is absolutely appropriate. First, they establish the protection ability of overexpression of dNMNAT in severed neurons. Their results are somewhat different from a previous study from the Freeman lab, but they propose that the stronger WD protection that they observe is likely to the fact that they used an untagged dNMNAT transgene (in contrast to myc-tagged), as before, or because of higher expression. While they never proved this, I don't think it's important to address. Then they also developed a new model for WD, of the JO^+ neurons - this is an excellent choice as it allowed them to assay for synaptic conservation (function of the circuit - *i.e.* antennal grooming). Next, they decided to perform an analysis of the translating transcriptome, using the TRAP technique. In short - they expressed GFP tagged Ribosome subunits and then isolated bound transcripts from severed axons. This allowed them to identify genes that are likely translated in axons and upregulated following injury. In figure 3, the authors develop a very cool automatic screen to measure optogenetic-induced grooming before and after injury and upon KD of candidate genes. Figures 4-5 focus on the results from that screen that identify genes involved in ubiquitination, calcium homeostasis, as well as TOR, as required for optogenetic-induced grooming following injury. I think the study demonstrated an impressive tour-de-force and on route generation of new tools and methods. As such, I support publication in EMBO reports pending the clarifications/corrections of my concerns as detailed below.

By far the weakest figures are #4+#5. The fact that the transcript were identified in a TRAP experiments using severed axons (*i.e.* translating transcripts are present in the axons), and that their KD using RNAi (throughout the cell, including the soma, and throughout

development including pre- and post-injury) does NOT prove, in my opinion, that their requirement for synaptic maintenance DEPENDS on their translation in axons. Unless I am missing something - there is no easy way around this experimentally. Thus, I would opt to change the text to more careful wording and use the discussion to promote the hypothesis that is spelled out (prematurely, in my opinion) at the very end of the results.

We agree with Referee #1; the *in vivo* experimental approach is fairly challenging to demonstrate a dependency on the local translation of these candidates. The evidence for our belief that local translation of these candidates is required to sustain functional preservation starts with isolating mRNAs bound to ribosomes one week after injury.

To further support this hypothesis, we employed experiments with Ricin, a genetic tool that inactivates ribosomes in a temperature-dependent manner. At 18 °C, Ricin is unfolded and inactive, whereas at 30 °C, it is properly folded and inactivates ribosomes and thus translation (Chen et al., 2012).

When we used 30 °C 24 h before axotomy (and until 7 dpa) to get active Ricin in *dnmnat^{OE}*, it already resulted in a reduced grooming phenotype before axotomy (Figure A, below). However, when we switched temperatures specifically after axotomy, the observed effect of antennal grooming was gone (Figure B, below). These results indicate that the temperature-sensitive Ricin tool is not useful for our designed experiments.

We also employed pharmacological inhibition of translation by feeding flies with 35 mM cycloheximide for 16 h (Zamurrad et al., 2018), which also changes behavior (Eschment et al., 2020). We decided to implement the protocol in our approach. However, we want to clarify that these experiments cannot be performed in a tissue-specific manner (e.g., see Figure 3D, E in (Zamurrad et al., 2018)). Feeding flies with cycloheximide affects probably all tissues in flies, including glia and neurons. Nevertheless, we exposed flies in various conditions to 35 mM cycloheximide after injury. First, we tested a 16 h feeding protocol ± 6 h starvation prior to 7 dpa (Figure A, B, below). However, the flies treated with 35 mM cycloheximide died. We repeated a similar experiment with 16 h feeding protocol ± 6 h starvation prior to 3 dpa (Figure C, D, below). Even under these conditions, the flies treated with 35 mM cycloheximide died. Lastly, we reduced the cycloheximide

concentration to 5 mM for 24 h, and were able to measure grooming at 7 dpa, since the flies survived (Figure E, F). While 5 mM cycloheximide is less toxic, it is important to note that it does not substantially block protein synthesis in wild-type flies (Bolduc et al., 2008). Our observations supported these findings, as antennal grooming was not affected by a 24 h 5 mM cycloheximide treatment.

We understand the concern raised by Referee #1: “The fact that the transcripts were identified in a TRAP experiments using severed axons, and that their KD using RNAi does NOT prove, in my opinion, that their requirement for synaptic maintenance DEPENDS on their translation in axons.”. We therefore toned down our conclusion in the result section for the protein ubiquitination results after Figure 4:

Our RNAi-mediated approach suggests that the identified transcripts by TRAP involved in protein ubiquitination are required to sustain *dnmna1^{OE}*-mediated synaptic function one week after axotomy.

Similarly, we toned down our conclusion in the result section after the calcium results in Figure 4:

Our observation provides further evidence that the transcripts involved in Ca^{2+} identified by TRAP are crucial for *dnmna1^{OE}*-mediated preservation of synaptic function.

And we toned down our conclusion in the result section after the mTOR findings in Figure 5:

Our observations support the notion that the identified transcripts by TRAP may be locally translated in severed axons and their synapses under the control of mTOR signaling to sustain their function.

We removed the last sentence in the result section, and promoted our hypothesis in the discussion as indicated below:

Based on our observations, we propose that severed projections with attenuated programmed axon degeneration employ local protein synthesis through mTOR signaling. Among the translated transcripts are candidates that help to cope with the turnover of already translated polypeptides by polyubiquitination and Ca^{2+} buffering. This model is supported by the impairment of either local translation or Ca^{2+} transport and polyubiquitination candidates, resulting in reduced synaptic function.

Altogether, our findings support the hypothesis of the “autonomous axon” (Alvarez, 2001), where local protein synthesis, with a pool of mRNAs, ensures the continued functional adjustments of projections where a nucleus is far away or the soma cut-off. In some insects, the soma is entirely absent due to the selective evolutionary pressure of brain miniaturization, where anucleate axons continue to

contribute to the behavior for the life span (Polilov, 2017, 2012). Interestingly, severed axons persist for weeks to months in various invertebrates and some vertebrates (Bittner, 1991, 1988). Here, local translation could ensure sustained maintenance of synaptic plasticity, which we observe in our model of impaired Wallerian degeneration.

Second - and perhaps most importantly - one of the most important new claims here (apart from the tools/methods developed - which merit publication by themselves), is that axon preservation and synaptic preservation are not the same thing. i.e. they claim that while local translation is not required for NMNAT inhibition of axonal WD (as was previously shown - but not demonstrated in their own hands) but is required for NMNAT inhibition of synaptic elimination during WD. There is a crucial, yet missing, experiment that needs to prove this. They need to show that while the antennal grooming is NOT preserved upon KD of selected genes (most notably Nckx30C, Raptor and S6K), the degeneration of the axonal projections IS inhibited.

We appreciated the suggestion and added now a whole figure (Appendix Figure S3) to demonstrate that the severed JO^+ projections appear unchanged, based on GFP fluorescence. We also quantified the brains. However, due to the large-scale approach, we assess the preservation of the projections as "intact" or "degenerated" as indicated in the bottom right corner of each example. Overall, we did not observe differences as compared to the controls. We added the following text to the result section for the protein ubiquitination, Ca^{2+} , and mTor candidates:

Importantly, among the three phenotypes, the morphology of the projections harbored no overt signs of degeneration (Figure 4B, Appendix Figure S3A).

[...]

The severed projections appeared similarly preserved compared to their controls (Figure 4C, Appendix Figure S3B).

[...]

As previously observed, the severed projections remained morphologically preserved (Figure 5B, Appendix Figure S3C).

Apart from that - the paper is well written and compelling. Some minor textual/presentation suggestions (and completely up to the authors/editor): 1) in the introduction it sounds as if WD is regulated only by 4 genes - this is the current state and

adding the words 'so far' would make it clearer. 2) two schemes, describing the wing injury experiment (otherwise, 1cb is quite cryptic to the reader who is unfamiliar with the system), and the grooming experiment (1G) - would help. 3) the very last sentence in the results would benefit from more accurate punctuation (I also suggest to move it to the discussion, as discussed above) - at the moment it is written as a causal chain of events, but there is no proof for that.

Minor point 1) We changed the introduction as indicated below:

In *Drosophila*, so far, programmed axon degeneration is mediated by four genes and a single metabolite (Fang et al., 2012; Llobet Rosell et al., 2022; Neukomm et al., 2017; Osterloh et al., 2012; Paglione et al., 2020; Xiong et al., 2012).

Minor point 2) In Figure 1, we added the schematic illustrations for all assays. We also changed the Figure legend accordingly.

Minor point 3). We removed the last sentence in the result section and implemented it in the discussion, as indicated below:

Based on our observations, we propose that severed projections with attenuated programmed axon degeneration employ local protein synthesis through **mTOR signaling**. Among the translated transcripts are candidates that help to cope with the turnover of already translated polypeptides by polyubiquitination and **Ca²⁺** buffering. This model is supported by the impairment of either local translation or **Ca²⁺** transport and polyubiquitination candidates, resulting in reduced synaptic function.

Referee #2:

The NAD-dependent axon destruction program is a relatively newly discovered and evolutionarily conserved biological pathway that is of broad interest not only to the community of molecular biology and neurobiology but also to clinical scientists, because its prevention (for example, by Sarm1 inhibition or Nmnat2 over-expression) may delay neurodegenerative diseases.

In this manuscript, Paglione and colleagues report the possibility of locally synthesized proteins, such as huwei, CG10916, nckx30C, and s6k, support synaptic function in severed, Wallerian degeneration (WD)-delayed axons in *Drosophila*. They generated UAS-GAL4 fly lines, in which dnmnat is over-expressed in GAL4-expressing neuronal populations. They

showed that *dnm1at* over-expression prevents WD at least for two weeks in four different GAL4 lines and that optogenetic activation of WD-delayed, severed axons elicits antennal grooming behavior, a proxy of intact synaptic transmission, in one line (JO^+ neurons), which they detect using the automated image analysis method they developed. For biochemical purification of ribosome-bound mRNAs from severed axons, they used the *orco*-GAL4 line that labels more neurons and axons and found that some mRNAs are enriched in axons after axotomy. RNAi-mediated depletion of some of those mRNAs in JO^+ neurons prevented the preservation of antennal grooming behavior after axotomy, suggesting that local translation of those mRNAs may contribute to the sustained synaptic transmission after axotomy.

This is a well-performed study reporting several interesting findings, tools, and datasets that may be of interest to the community of molecular biology and neurobiology. The manuscript is written in an engaging way and the discussion is comprehensive. I have the following comments that the authors may consider in making this study more accessible to readers.

<Major comments>

1. The requirement of local translation in synaptic function

Blocking axon destruction program delays WD, and after such manipulations, severed axon terminals retain intact morphology and function. How synaptic function is maintained over weeks without the supply of new materials from the soma is a key question in the field. The most important finding of this manuscript, in this sense, is perhaps that proteins synthesized in severed and WD-delayed axons support synaptic function over weeks.

The authors used TRAP in the first part of the manuscript to identify mRNAs whose axonal translation increases after injury. In the latter part of the manuscript, the authors used RNAi-mediated knockdown of the candidate mRNAs to investigate the requirement of these genes in synaptic function. The authors interpreted these results as some of the mRNAs on which they performed functional analyses support synaptic function by local translation.

Although this interpretation is logical, the data do not directly show that local translation of these mRNAs is required for sustained synaptic transmission, because RNAi was performed before axotomy and in the entire neuron including the soma.

While the study suggests that locally synthesized proteins support synaptic function in severed axons, the interpretation of the RNAi-mediated knockdown experiments should be cautious. Conducting knockdown experiments specifically in axons or after axotomy in *dnm1at*^{OE} neurons would strengthen the evidence for the requirement of local translation. If this is not feasible, the authors should tone down the statement such as the last sentence

of page 10 ("Thus, our RNA-mediated approach provides evidence that the local translation of ...") and qualify the interpretation of these experiments.

We agree with Referee #2 on: "*Conducting knockdown experiments specifically in axons or after axotomy in *dnmnat*^{OE} neurons would strengthen the evidence for the requirement of local translation.*" We employed pharmacological inhibition of translation by feeding flies with 35 mM cycloheximide for 16 h (Zamurrad et al., 2018), which also impacts on their behavior (Eschment et al., 2020). We decided to implement the protocol in our approach. However, we want to clarify that these experiments cannot be performed in a tissue-specific manner (e.g., see Figure 3D, E in (Zamurrad et al., 2018)). Feeding flies with cycloheximide affects probably all tissues in flies, including glia and neurons. Nevertheless, we exposed flies in various conditions to 35 mM cycloheximide after injury. First, we tested a 16 h feeding protocol ± 6 h starvation prior to 7 dpa (Figure A, B, next page). However, the flies treated with 35 mM cycloheximide died. We repeated a similar experiment with 16 h feeding protocol ± 6 h starvation prior to 3 dpa (Figure C, D, below). Even under these conditions, the flies treated with 35 mM cycloheximide died. Lastly, we reduced the cycloheximide concentration to 5 mM for 24 h, and were able to measure grooming at 7 dpa, since the flies survived that treatment (Figure E, F). It is important to note that 5 mM cycloheximide is less toxic, but it does not substantially block protein synthesis in wild-type flies (Bolduc et al., 2008). Our observations supported these findings, as antennal grooming was not affected by a 24 h 5 mM cycloheximide treatment.

We therefore toned down our conclusion in the result section for the protein ubiquitination results after Figure 4:

Our RNAi-mediated approach **suggests that the identified transcripts by TRAP involved in protein ubiquitination are required to sustain *dnmnat*^{OE}-mediated synaptic function one week after axotomy.**

Similarly, we toned down our conclusion in the result section after the calcium results in Figure 4:

Our observation provides **further evidence that the transcripts involved in Ca²⁺ identified by TRAP are crucial for *dnmnat*^{OE}-mediated preservation of synaptic function.**

And we toned down our conclusion in the result section after the mTOR findings in Figure 5:

Our observations support **the notion that the identified transcripts by TRAP may be locally translated in severed axons and their synapses under the control of mTOR signaling to sustain their function.**

We also moved the last sentence of the results section to, expanded our interpretation in the discussion:

Based on our observations, we propose that severed projections with attenuated programmed axon degeneration employ local protein synthesis through **mTOR signaling**. Among the translated transcripts are candidates that help to cope with the turnover of already translated polypeptides by polyubiquitination and **Ca²⁺** buffering. This model is supported by the impairment of either local translation or **Ca²⁺** transport and polyubiquitination candidates, resulting in reduced synaptic function.

Altogether, our findings support the hypothesis of the “autonomous axon” (Alvarez, 2001), where local protein synthesis, with a pool of mRNAs, ensures the continued functional adjustments of projections where a nucleus is far away or **the soma cut-off**. **In some insects, the soma is entirely absent due to the selective evolutionary pressure of brain miniaturization, where anucleate axons continue to contribute to the behavior for the life span** (Polilov, 2017, 2012). **Interestingly, severed axons persist for weeks to months in various invertebrates and some vertebrates** (Bittner, 1991, 1988). **Here, local translation could ensure sustained maintenance of synaptic plasticity, which we observe in our model of impaired Wallerian degeneration.**

2. Possible new experiments

Because the main message of this manuscript is the local translation in pre-synaptic function, I think that alternative functional experiments could be performed. For example, the authors could perform time-course experiments on antennal grooming behaviors shortly after axotomy.

We performed a time-course experiment after injury with a knockdown of *s6k* in *dnmnat^{OE}* compared to *dnmnat^{OE}*. We observed that the evoked antennal grooming behavior of *dnmnat^{OE}* did not change over time ($R^2 = 0.06236$, with $p = 0.0742$), while *s6k^{RNAi}* resulted in a significant decrease ($R^2 = 0.2490$, with $p = 0.0006$). We added our observations to Figure 5C, updated the figure legend and the results section accordingly:

To support the *s6k^{RNAi}* findings, we performed a time-course analysis where evoked antennal grooming was measured at 1, 3, 5, and 7 dpa in the same animal (Figure 5C, Appendix Figure S4). We observed that the evoked antennal grooming

behavior of *dnmnat*^{OE} did not change over time ($R^2 = 0.06236$, with $p = 0.0742$), while *s6k*^{RNAi} resulted in a significant decrease ($R^2 = 0.2490$, with $p = 0.0006$).

Repeated (or continuous) optogenetic stimulation on the day of axotomy will lead to depletion of pre-synaptic materials and premature cessation of stimulation-induced antennal grooming behavior (perhaps on the same day of axotomy). Recovery of the behavior will indicate replenishment of presynaptic materials, and if it happens by local translation, it may be possible to block it either by cycloheximide or RNAi-mediated depletion of the effective genes (e.g., *huwei*, *cg10916*, *nckx30C*, and *s6k*).

Repeated or continuous optogenetic stimulation was already engaged aiming to deplete presynaptic pools of the readily releasable pool (RRP) of neurotransmitter-filled vesicles (Watanabe et al., 2013). Within seconds, the RRP would be depleted, but ultrafast endocytosis ensures rapid recycling of the RRP.

Importantly, in our assay, we stimulate severed *dnmnat*^{OE} axons 3 times, with 10 Hz for 10 s with a 30 s recovery between stimulations (Figure 3B). The accumulated stimulation does not alter the grooming behavior during each stimulation, arguing that despite the long and non-physiological stimulation, synapses are not depleted from neurotransmitter-filled vesicles. Moreover, repeated evoked neurotransmitter release by optogenetics can result in off-target neuronal activation and damage (Guo et al., 2022). In our hands, prolonged exposure resulted in animal lethality. Therefore, we avoided experiments with repeated or continuous optogenetic stimulation. Nevertheless, we were successful with the time course where animals are exposed to red-light stimuli at 1, 3, 5, and 7 dpa, as discussed above.

3. The results of TRAP experiments

The results of DEG experiments should be provided as supplemental tables, and whenever possible the gene names or symbols should be annotated in figures (e.g., Figure S3).

We added the DEG experiments providing the DESeq2 output analysis as supplementary Datasets. We also added the name of the genes in Figure 2D, and in Figure S3 which is now Expanded View Figure EV3.

<Minor comments>

4. Figure 1A

1) Both upper and lower panels are labeled with "1 cb", but it is difficult to see where the cell bodies are located in the figure.

To better understand the assay, in Figure 1, we added the schematic illustrations for all assays. We also changed the Figure legend accordingly.

2) In the wild type, one axon appears intact even 14 days after axotomy. Are some of the *Dpr1*⁺ neurons resistant to Wallerian degeneration?

We apologize for the confusion. We added a schematic illustration in Figure 1 that explains our axotomy protocol. Sometimes, we can observe uninjured control axons (one axon from one uninjured neuronal cell body) in the field of view. It depends on where exactly the wing is cut. We also changed the Figure legend accordingly.

5. Figure 2A

It is difficult to appreciate where the cell bodies and axon terminals are located in the diagrams of the head. I understand that the large grey circle inside the head indicates the antennal lobe, which does not contain the cell bodies of *orco*⁺ neurons. But the small green circles may be mistaken for the cell bodies. The authors should annotate the diagrams to clarify the head contains no cell bodies of the labeled neurons.

In Figure 1H, we specified the cell bodies, axons, and synaptic terminals in the schematic illustration. We also changed the Figure legend accordingly.

6. Background group in TRAP

The authors used the *GAL4*-only group as the background control. It is not specified whether axotomy was performed and whether TRAP was performed (if so, when TRAP was performed in relation to axotomy).

For clarification: in all our experiments, we used adult heads, TRAP-seq, and library preparation. For this reason, we applied the same type of normalization to the three genotypes. In Figure EV4, we aimed to identify the most significantly enriched biological classes. We added TRAP below the arrow in Figure EV4A, and in the results section indicated below:

We **also** implemented an additional control using **TRAP-seq from *orco-Gal4* alone (background)** to further restrict our analyses (**Figure EV4A**).

7. Polyribosomes

Unless the authors have evidence that what they pulled down by TRAP are polyribosomes, the term should be replaced with ribosomes.

In the results section, we removed the polysomes as indicated below:

Subsequently, axonal and synaptic-specific ribosomes were immunoprecipitated with anti-GFP antibody-coated magnetic beads, mRNAs extracted, and reverse transcribed to establish cDNA libraries (Figure 2A, Material and Methods).

8. Discussion

The Discussion section is interesting but has paragraphs that are not directly relevant to the findings of the manuscript (such as those on Rabies viruses and hibernating vertebrates). On the contrary, the discussion on the genes whose RNAi-mediated knockdown leads to defective development/assembly of synapses (such as *traf* and *cac*) or the maintenance of synaptic function after axotomy (such as *huwei* and *s6k*) is missing. The authors should expand the discussion on points that are relevant to their findings.

We appreciate the suggestion of removing the Rabies virus and hibernating vertebrates section in the discussion. We changed the discussion as indicated below:

Highly translated mRNAs are selectively degraded in axons; thus, their pools decrease over time (Jung et al., 2023). We used an RNAi-based screen to lower the neuronal mRNA pool. Before injury, in projections, the pool may offer sufficient mRNA substrates for local translation. However, 7 days post axotomy, the pool may be below the threshold where local translation is significantly reduced. There, it could result in reduced grooming behavior if the candidates are involved in sustaining synaptic function.

Interestingly, *traf4*^{RNAi} and *cac*^{RNAi} reduced grooming in non-axotomized animals. Since their projections appeared unaffected, it suggests that *traf* and *cac* harbor a more general function in neuronal communication. In contrast, we identified candidates that resulted in reduced grooming solely after axotomy. Lowering their mRNA pools did not affect grooming in non-axotomized animals but resulted in reduced grooming specifically after axotomy. Since neuronal morphology remained unaffected, it is tempting to speculate that such candidates are locally translated to sustain synaptic function. Among them are genes in protein ubiquitination (*huwe1*, *CG10916*, and *CG6923*), and *nckx30c* involved in Ca²⁺ homeostasis. Remarkably,

we also observed increased grooming solely after axotomy in *calx^{RNAi}* animals. Further studies are required to dissect the underlying mechanism..

[...]

Based on our observations, we propose that severed projections with attenuated programmed axon degeneration employ local protein synthesis through **mTOR signaling**. Among the translated transcripts are candidates that help to cope with the turnover of already translated polypeptides by polyubiquitination and **Ca²⁺** buffering. This model is supported by the impairment of either local translation or **Ca²⁺** transport and polyubiquitination candidates, resulting in reduced synaptic function.

Altogether, our findings support the hypothesis of the “autonomous axon” (Alvarez, 2001), where local protein synthesis, with a pool of mRNAs, ensures the continued functional adjustments of projections where a nucleus is far away or **the soma cut-off**. **In some insects, the soma is entirely absent due to the selective evolutionary pressure of brain miniaturization, where anucleate axons continue to contribute to the behavior for the life span (Polilov, 2017, 2012). Interestingly, severed axons persist for weeks to months in various invertebrates and some vertebrates (Bittner, 1991, 1988). Here, local translation could ensure sustained maintenance of synaptic plasticity, which we observe in our model of impaired Wallerian degeneration.**

9. Incomplete sentence

The sentence starting with "In *JO⁺ dnmnat^{OE}* neurons, RNAi-mediated" on page 11 appears to be incomplete.

We changed the sentence by removing the word "injury", as indicated below:

In *JO⁺ dnmnat^{OE}* neurons, *raptor^{RNAi}* decreased behavior in uninjured controls, and at 7 dpa, we observed a statistically non-significant trend of a further decrease (Figure 5B).

10. Citation

"ZenXian Nious, 2002" is formatted in a different way than the other references.

We changed the reference accordingly.

Referee #3:

Evidence, reproducibility and clarity:

In the present study, Paglione *et al.* describe the effects of local translation on programmed axon degeneration (Wallerian degeneration) in a *Drosophila* model of axotomy. They utilise a dNmnat overexpression system to preserve injured axons and identify transcripts that differentially interact with ribosomes when the NAD⁺ homeostasis is perturbed. Here, they find transcripts encoding proteins with key roles in mTORC1 pathway, protein ubiquitination and calcium homeostasis, a similar set of biological processes dysregulated in mammalian models of Wallerian degeneration. They also present a novel high-throughput behaviour assay tool to assess synaptic activity in the JO⁺ circuit, which was validated with an impressive set of control experiments. Knockdowns of preferentially translated mRNAs identified from the dNmnat OE system lead to reduced level of induced grooming behaviour, consistent with the reversal of programmed axon degeneration to the 'normal' rate. Overall, the key conclusions of the paper align with the conducted experiments, but a few additional experiments or clarification of the methods may be necessary to provide more robust conclusions.

Significance:

This work offers novel insights into the process of axon preservation although its molecular mechanism of action is not well characterised. The authors identify a group of locally translated and evolutionary-conserved transcripts that respond to altered NMD/NAD⁺ ratio, which has been shown to trigger Wallerian degeneration. Together with the deep learning-aided grooming behaviour paradigm, this work will be valuable in dissecting the precise mechanism of pathological axon degeneration/preservation.

Major comments:

1) In Figure 2D, I don't think the experiment supports the claim "We identified transcripts of protein ubiquitination enriched in *dnmnat*^{OE} solely after injury". Here the authors are comparing translation profiles of <entire neuron> vs <neurites>. This can be a problem because the translation level in cell bodies could overwhelm the level in neurites, as the authors mentioned in Page 6. A proper comparison should be set in the same sub-compartment e.g. <non-ablated condition neurites> vs , or should be limited to the same compartments. Comparison of whole-neuron translome between wildtype and dNmantOE, however, is very useful.

We apologize for the confusion, which indicates that we failed to explain how we performed our analyses. We did not perform a two-paired comparison. For clarification, in Figure 2D and Figure EV3, we did not compare transcripts from the uninjured neurons (before injury) with severed axons (after injury), as it may have been understood.

Instead, what we did, first, we compared wild type and *dnmnat*^{OE} after injury (Figure 2B). We used the 499 enriched transcripts from *dnmnat*^{OE} to perform a GO term analysis. These biological process GO terms are shown in Figure 2C.

In Figure 2D, for example, under protein ubiquitination, we used all significantly enriched classes, namely: *protein ubiquitination* (GO:0016567) and *protein ubiquitination involved in ubiquitin-dependent protein catabolic process* (GO:0042787). They were combined to generate a novel gene list (to not eliminate potential candidates, but removing existing duplicates). We used the resulting rLog data to generate a heatmap to check how those genes are enriched in different conditions (e.g., control and 7 dpa, respectively) and genotypes (e.g., wild type and *dnmnat*^{OE}, respectively). We repeated this analysis for each GO term class in Figure 2C. Based on this analysis, we found that protein ubiquitination genes were the sole class enriched after injury in *dnmnat*^{OE}.

However, we also performed the comparison between wild type and *dnmnat*^{OE} before injury (Figure A, below) as requested by the reviewer. And we also added the GO term classes enriched in *dnmnat*^{OE} (Figure B, below). However, we feel that this comparison is not the major goal of our project. The data is accessible in the repository, and scientists may use it to perform additional analyses.

We changed the results section indicated below:

Next, we looked at the expression levels (e.g., rLog values) of our 1033 identified GO term class transcripts (Figure 2C) in the distinct conditions and genotypes (e.g., control, 7 dpa, wild type, and *dnmnat*^{OE}, respectively) (see Materials and Methods for details). Interestingly, protein ubiquitination transcripts were the sole GO term class enriched in *dnmnat*^{OE} solely after axotomy (Figure 2D). In contrast, transcripts of Ca²⁺ transport were ...

We also changed the text in Materials and methods indicated below:

Gene Ontology (GO) term enrichment analysis for biological processes was performed with the Database for Annotation, Visualization, and Integrated Discovery (DAVID) 6.8 (Huang et al., 2009). Statistical analyses and data visualization were conducted in R using the base, dplyr, pheatmap, and ggplot2 packages.

In Figure 2D and EV3, the following procedure was applied to each class: for example, under protein ubiquitination, significantly enriched classes were used to avoid eliminating potential candidates: protein ubiquitination (GO:0016567) and protein ubiquitination involved in ubiquitin-dependent protein catabolic process (GO:0042787). The rLog data from the newly generated gene list was used to generate a heatmap to assess transcript enrichment in different conditions (e.g., control and 7 dpa, respectively) and genotypes (e.g., wild type and *dnmna1^{OE}*, respectively). The analysis was repeated for each GO term class (Figure 2C).

2) In Page 8, the additional background filtering method should be better clarified. Is the background control library (*orco-Gal4*) an RNA-seq library? Was the entire adult head used for the library preparation? Comparing RNA-seq and TRAP-seq libraries often requires a different normalisation approach, which should be described in the Methods. If the aim was to filter out transcripts with low/non-reproducible readcounts, why not use a simple transcripts per million cut-off from TRAP-seq?

For clarification: in all our experiments, we used adult heads, TRAP-seq, and library preparation. For this reason, we applied the same type of normalization to the three genotypes. In Figure EV4, we aimed to identify the most significantly enriched biological classes. We added TRAP below the arrow in Figure EV4A, and in the results section indicated below:

We **also** implemented an additional control using **TRAP-seq from *orco-Gal4*** alone (background) to further restrict our analyses (**Figure EV4A**).

3) In Figure 4, can the authors show images of accelerated axon degeneration (as in Figure 1) in conditions where grooming behaviours are affected? The morphological data will nicely complement the behavioural information.

We appreciated the suggestion and added now a whole figure (Appendix Figure S3) to demonstrate that the severed JO^+ projections appear unchanged, based on GFP fluorescence. We also quantified the brains. However, due to the large-scale approach, we assess the preservation of the projections as “intact” or “degenerated” as indicated in the bottom right corner of each example. Overall, we did not observe differences as compared to the controls. We added the following text to the result section for the protein ubiquitination, Ca^{2+} , and mTor candidates:

Importantly, among the three phenotypes, the morphology of the projections harbored no overt signs of degeneration (Figure 4B, Appendix Figure S3A).

[...]

The severed projections appeared similarly preserved compared to their controls (Figure 4C, Appendix Figure S3B).

[...]

As previously observed, the severed projections remained morphologically preserved (Figure 5B, Appendix Figure S3C).

Minor comments:

1) Tables of TRAP-seq DESeq2 output and DAVID gene ontology analysis should be included as supplementary files.

We added the DESeq2 output and DAVID gene ontology analysis as supplementary Datasets.

2) TRAP-seq dataset should be uploaded to public repositories (e.g. Gene Expression Omnibus).

We uploaded the TRAPseq data to the Gene Expression Omnibus (reviewer code: otajmiwqjtqjrgp, dataset code: GSE270011), which remains private until August 31st, 2025, which will be changed to public once accepted for publication:

<https://www.ncbi.nlm.nih.gov/geo/query/acc.cgi?acc=GSE270011>

3) In Figure S3, it would be helpful to show the name of the genes next to the heatmaps since only a handful of genes are displayed per GO category.

We added the name of the genes in Figure 2D, and in Figure S3 which is now Expanded View Figure EV3.

4) TRAP-seq was performed in the *Orco*⁺ system while the behavioural assay was done in *JO*⁺ neurons. Can the authors comment/discuss conservation of axon degeneration pathways in neuron subtypes? Are the genes with no RNAi behavioural phenotype expressed in both systems?

Programmed axon degeneration is evolutionarily conserved across species, as indicated in the introduction (Llobet Rosell and Neukomm, 2019). In addition, the core proteins (e.g., MYCBP2, NMNAT2 and SARM1 in mammals / Highwire, dNmnat and dSarm in *Drosophila*), are pan-neuronally expressed. In addition, we found a protective *dnmnat*^{OE} role in cholinergic *orco*⁺ chordotonal neurons in antennae and glutamatergic/cholinergic chemosensory neurons in the wing (Neukomm et al., 2014; Salvaterra and Kitamoto, 2001).

Flybase offers RNAseq data from the Fly Cell Atlas, where expression data from *orco*⁺ and *JO*⁺ neurons are indicated. We found that both *orco*⁺ and *JO*⁺ neurons expressed the candidate genes (with and without a behavioral phenotype). We changed the text in the results section indicated below:

The comparisons of the mammalian dataset with our background filtering strategy prompted us to validate protein ubiquitination and Ca²⁺ transport GO term classes (Figure 2E, F, Figure EV4E, F). The expression of candidate genes overlapped among *orco*⁺ and *JO*⁺ sensory neurons, supporting our subsequent validation in *JO*⁺ neurons.

5) In Figure 4, are the RNAi lines used for the knockdown experiments validated?

There is a continuous effort in the *Drosophila* community to validate and improve RNAi lines (Perkins et al., 2015). To facilitate large-scale functional studies in *Drosophila*, the *Drosophila* Transgenic RNAi Project (TRiP) at Harvard Medical School (HMS) was established along the Bloomington *Drosophila* Stock Center (BDSC), with several goals: developing efficient vectors for RNAi that work in all tissues, generating a genome-scale collection of RNAi stocks with input from the community, distributing the lines as they are generated through existing stock centers, validating as many lines as possible using RT-qPCR and phenotypic analyses, and developing tools and web resources for identifying RNAi lines and retrieving existing information on their quality (e.g., off-target effects or knockdown efficiency). Data on the characterization of the lines either by RT-

qPCR or phenotype is available on a dedicated website, the RNAi Stock Validation and Phenotypes Project (RSVP, <http://www.flyrnai.org/RSVP.html>).

6) In the Method section regarding mouse/fly orthologs, how was the high rank assessed? Did the authors use a particular DIOPT score threshold?

We used the high-rank DIOPT, which indicates that both forward and reverse searches resulted in the best score with a DIOPT score ≥ 2 . We changed the text in Materials and methods indicated below:

Only orthologs with high rank, best score, and DIOPT score ≥ 2 were chosen to compare with our data set.

References

- Bolduc, F.V., Bell, K., Cox, H., Broadie, K.S., Tully, T., 2008. Excess protein synthesis in *Drosophila* Fragile X mutants impairs long-term memory. *Nat. Neurosci.* 11, 1143–1145. <https://doi.org/10.1038/nn.2175>
- Chen, C.-C., Wu, J.-K., Lin, H.-W., Pai, T.-P., Fu, T.-F., Wu, C.-L., Tully, T., Chiang, A.-S., 2012. Visualizing Long-Term Memory Formation in Two Neurons of the *Drosophila* Brain. *Science* 335, 678–685. <https://doi.org/10.1126/science.1212735>
- Eschment, M., Franz, H.R., Güllü, N., Hölscher, L.G., Huh, K.-E., Widmann, A., 2020. Insulin signaling represents a gating mechanism between different memory phases in *Drosophila* larvae. *PLoS Genet.* 16, e1009064. <https://doi.org/10.1371/journal.pgen.1009064>
- Guo, J., Wu, Y., Gong, Z., Chen, X., Cao, F., Kala, S., Qiu, Z., Zhao, X., Chen, J., He, D., Chen, T., Zeng, R., Zhu, J., Wong, K.F., Murugappan, S., Zhu, T., Xian, Q., Hou, X., Ruan, Y.C., Li, B., Li, Y.C., Zhang, Y., Sun, L., 2022. Photonic Nanojet-Mediated Optogenetics. *Adv. Sci.* 9, 2104140. <https://doi.org/10.1002/adv.202104140>
- Llobet Rosell, A., Neukomm, L.J., 2019. Axon death signalling in Wallerian degeneration among species and in disease. *Open Biol* 9, 190118. <https://doi.org/10.1098/rsob.190118>
- Neukomm, L.J., Burdett, T.C., Gonzalez, M.A., Zuchner, S., Freeman, M.R., 2014. Rapid in vivo forward genetic approach for identifying axon death genes in *Drosophila*. *Proc Natl Acad Sci USA* 111, 9965–9970. <https://doi.org/10.1073/pnas.1406230111>
- Perkins, L.A., Holderbaum, L., Tao, R., Hu, Y., Sopko, R., McCall, K., Yang-Zhou, D., Flockhart, I., Binari, R., Shim, H.-S., Miller, A., Housden, A., Foos, M., Randkelv, S., Kelley, C., Namgyal, P., Villalta, C., Liu, L.-P., Jiang, X., Huan-Huan, Q., Wang, X., Fujiyama, A., Toyoda, A., Ayers, K., Blum, A., Czech, B., Neumuller, R., Yan, D., Cavallaro, A., Hibbard, K., Hall, D., Cooley, L., Hannon, G.J., Lehmann, R., Parks, A., Mohr, S.E., Ueda, R., Kondo, S., Ni, J.-Q., Perrimon, N., 2015. The Transgenic RNAi Project at Harvard Medical School: Resources and Validation. *Genetics* 201, 843–852. <https://doi.org/10.1534/genetics.115.180208>
- Salvaterra, P.M., Kitamoto, T., 2001. *Drosophila* cholinergic neurons and processes visualized with Gal4/UAS–GFP. *Gene Expr. Patterns* 1, 73–82. [https://doi.org/10.1016/s1567-133x\(01\)00011-4](https://doi.org/10.1016/s1567-133x(01)00011-4)
- Watanabe, S., Rost, B.R., Camacho-Pérez, M., Davis, M.W., Söhl-Kielczynski, B., Rosenmund, C., Jorgensen, E.M., 2013. Ultrafast endocytosis at mouse hippocampal synapses. *Nature* 504, 242–247. <https://doi.org/10.1038/nature12809>
- Zamurrad, S., Hatch, H.A.M., Drelon, C., Belalcazar, H.M., Secombe, J., 2018. A *Drosophila* Model of Intellectual Disability Caused by Mutations in the Histone Demethylase KDM5. *Cell Rep.* 22, 2359–2369. <https://doi.org/10.1016/j.celrep.2018.02.018>

Dear Lukas,

Thank you for the submission of your revised manuscript. We have now received the enclosed reports from the referees and I am happy to say that all support its publication now. Only a few editorial requests will need to be addressed before we can proceed with the official acceptance of your manuscript:

- The ms has 5 main figures but separate Results and Discussion sections. Please either add one more main figure or combine results and discussion to publish your ms as a short report with a maximum of 29.000 characters.
- Please add up to 5 keywords to the ms file.
- Please correct "Bibliography" to "References"
- Please add a Data Availability Section (DAS) to the end of the Methods that contains accession numbers and links to your publicly deposited data.
- Please add a "DISCLOSURE AND COMPETING INTERESTS STATEMENT"
- Please remove "data not shown" on page 24 as per journal policy.
- It would be good if the single figures would fit roughly on a single page.
- The legends for the Dataset files should be included as a separate tab in the excel files themselves.
- Please remove the Instructions and the Example from the Reagents and Tools Table.
- The movie files need to be renamed to Movie EV1-EV3 with the corresponding callouts, and the legends should be removed from ms file and zipped with each movie file.
- The ms section order should be corrected to: title page with complete author information, abstract, keywords, introduction, results, discussion, methods, data availability section, acknowledgements, disclosure and competing interests statement, references, main figure legends, tables, expanded figure legends.

Comments on the Figure Legends:

1. Please note that the exact p values are not provided in the legends of figures 1d, f-g, i; 4b-c; 5b; EV 1; EV 2b; EV 6.
2. Please indicate the statistical test used for data analysis in the legends of figures 2c, f; EV 4f.
3. Please note that the box plots need to be defined in terms of minima, maxima, centre, bounds of box and whiskers, and percentile in the legends of figures 1g; 3e, h; 4b-c; 5b; EV 1; EV 6.
4. Please note that information related to n is missing in the legends of figures 3c, f.
5. Please note that the error bars are not defined in the legends of figures 4c; 5b-c.
6. Please note that the white arrows are not defined in the legend of figure 1e, h. This needs to be rectified.

I would like to suggest a few changes to the title and abstract that needs to be written in present tense. Please address my comments below and let me know whether you agree with the following changes:

Local translation sustains synaptic function in impaired Wallerian degeneration

[Actually, based on the abstract your study does not seem to show that local translation is required to sustain synaptic function?]

After injury, severed axons separated from their somas activate programmed axon degeneration, a conserved pathway to initiate their degeneration within a day. Conversely, severed projections deficient in programmed axon degeneration remain morphologically preserved with functional synapses for weeks to months after axotomy. How this synaptic function is sustained remains currently unknown. Here, we show that dNmnat overexpression [OK??] attenuates programmed axon degeneration in distinct neuronal populations. Severed projections remain morphologically preserved for weeks. When evoked, they elicit a postsynaptic behavior, a readout for preserved synaptic function.

We used ribosomal pulldown to isolate translomes from these projections. Transcriptional profiling revealed several enriched biological classes [please re-write, it is not clear what you would like to say].

May be: Isolated translomes and transcriptional profiling reveals....

Identified candidates [candidates for what or of what??] were validated in a screen using an automated system to detect evoked antennal grooming as a proxy for preserved synaptic function. We identify mTOR as a mediator of protein synthesis in severed axons, and candidates involved in protein ubiquitination and Ca²⁺ homeostasis, required for preserved synaptic function.

EMBO press papers are accompanied online by A) a short (1-2 sentences) summary of the findings and their significance, B) 2-3 bullet points highlighting key results and C) a synopsis image that is exactly 550 pixels wide and 200-600 pixels high (the height is variable). The synopsis image should provide a sketch of the major findings, like a graphical abstract. Please note that text needs to be readable at the final size. Please send us this information along with the final manuscript.

Kind regards,
Esther

Referee #1:

I am happy with the response and changes that were implemented into the paper. I thought it was a super exciting paper even during the first submission - now it is much improved and should definitely be accepted as is...
Congratulations!

Referee #2:

The authors successfully addressed this reviewer's previous concerns, and I recommend the revised manuscript for publication in EMBO Reports.

Referee #3:

SUMMARY

The authors have addressed the concerns raised by the referees well, so the manuscript should be accepted for publication without further revision.

Re-evaluation of the revised manuscript

In the present study, Paglione et al. describe the effects of local translation on programmed axon degeneration (Wallerian degeneration) in a *Drosophila* model of axotomy. They utilise a dNm^{nat} overexpression system to preserve injured axons and identify transcripts that differentially interact with ribosomes when the NAD⁺ homeostasis is perturbed. In the revised manuscript, the authors have done well to justify their methodologies in their application of TRAP-seq to identify transcripts that have differential translational capacity during axon degeneration.

Key changes in the revised manuscript are:

Revision of the conclusion regarding the implication of local translation where the authors investigated the effects of RNAi knockdown of TRAP-identified transcripts.

Attempt to test the role of local translation using pharmacological inhibitions (Ricin, cycloheximide), but I do agree with the authors that it is not trivial to assess local translation specifically in axons and is beyond the scope of the current manuscript. Clarifications of the methodology where the authors compared TRAP-seq libraries between wild-type and dnm^{nat}OE samples. Clarification on the nature of the background control library (orco-GAL4), and how the sequencing libraries were normalised. Additional figure showing microscopes images of intact/degenerated axons that accompanies their large-scale grooming behaviour survey.

Provision of DESeq2 and DAVID GO output, as well as deposition of raw sequencing data on GEO.

Significance:

This work offers novel insights into the process of axon preservation via less-studied post-transcriptional regulation of gene expression control. Together with the deep learning-aided grooming behaviour paradigm, this work will be valuable in dissecting the precise molecular mechanism of pathological axon degeneration/preservation. Overall, the key conclusions of the revised manuscript now align well with the conducted experiments. This work may be of interest to neurobiology and molecular biology communities. The authors have addressed the concerns raised by the referees well, so the manuscript should be accepted for publication without further revision.

All editorial and formatting issues were resolved by the authors.

Prof. Lukas Neukomm
University of Lausanne
Department of Fundamental Neurosciences
Rue du Bugnon 9
Lausanne, Vaud 1005
Switzerland

Dear Lukas,

I am very pleased to accept your manuscript for publication in the next available issue of EMBO reports. Thank you for your contribution to our journal.

Best,
Esther
